# TopER: Topological Embeddings in Graph Representation Learning

## Abstract

Graph embeddings play a critical role in graph representation learning, allowing machine learning models to explore and interpret graph-structured data. However, existing methods often rely on opaque, high-dimensional embeddings, limiting interpretability and practical visualization.

In this work, we introduce Topological Evolution Rate (TopER), a novel, low-dimensional embedding approach grounded in topological data analysis. TopER simplifies a key topological approach, Persistent Homology, by calculating the evolution rate of graph substructures, resulting in intuitive and interpretable visualizations of graph data. This approach not only enhances the exploration of graph datasets but also delivers competitive performance in graph clustering and classification tasks. Our TopER-based models achieve or surpass state-of-the-art results across molecular, biological, and social network datasets in tasks such as classification, clustering, and visualization.

## 1 Introduction

Graphs are a fundamental data structure utilized extensively to model complex interactions within various domains, such as social networks (Leskovec et al., 2008), molecular structures (You et al., 2018), and transportation systems (Duan et al., 2022). Their inherent flexibility, however, introduces significant challenges when applied to machine learning tasks, primarily due to their irregular and high-dimensional nature. Due to the fact that graph data lacks inherent ordering and consistent dimensionality, traditional machine learning methods—designed for data in vector spaces—struggle with it.

Graph Neural Networks (GNNs) have emerged as the state-of-the-art models for tackling graph machine learning tasks due to their ability to learn effectively from graph-structured data. In the predominant paradigm of message-passing GNNs, the process begins by generating node embeddings. These embeddings can then be used in tasks such as node classification or link prediction. However, for graph-related tasks, such as molecular property prediction, the embeddings must be aggregated through a pooling layer to form graph-level representations. This method is computationally intensive, largely because the generation and management of node embeddings as intermediate steps substantially increase the overall computational burden. Ideally, an approach would allow for the direct creation of graph embeddings, circumventing the need to generate node-level representations first. Furthermore, these graph embeddings must be both low-dimensional and interpretable to maximize their practical utility and efficiency in various applications.

Topological Data Analysis (TDA) is well-suited for directly constructing graph representations without costly node embeddings. Topology studies the shape of data, and TDA primarily focuses on the qualitative properties of space, such as continuity and connectivity (Coskunuzer and Akçora, 2024). A particularly effective technique in TDA is *Persistent Homology (PH)*, which tracks topological features—like connected components and cycles—across various scales via a process known as filtration. Filtration is adept at revealing both local and global structures within graphs. It proves exceptionally useful for comparing graphs of different sizes that maintain the same inherent structure, which may suggest similar properties in graph datasets. For instance, similar substructures in protein interaction networks across different species may indicate comparable biological functions. By focusing on data shape, PH proves invaluable in graph tasks that benefit from a graph-centric approach, offering insights that might not be as apparent when focusing on individual node analysis. This shift from a node-centric to a graph-centric perspective can dramatically improve the understanding and

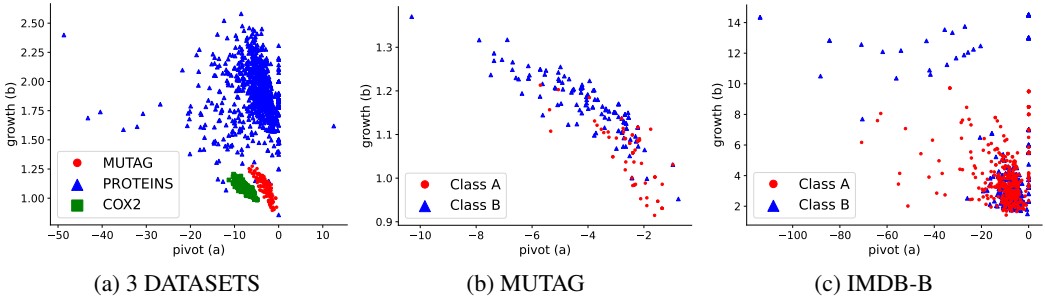

(a) 3 DATASETS  (b) MUTAG  (c) IMDB-B

Figure 1: **TopER Visualizations.** In the figures above, each point represents an individual graph. On the left, TopER is applied to three benchmark compound datasets using closeness sublevel filtration. The middle panel zooms in on the red point cloud from the left, demonstrating TopER's effectiveness in distinguishing between classes within the MUTAG dataset. On the right, a TopER visualization for the IMDB-B dataset is displayed.

application of graph data in fields like bioinformatics and network analysis. However, the utility of Persistent Homology is limited by the high computational demands involved in extracting topological features during the filtration process, mainly due to its cubic time complexity (Otter et al., 2017). This constraint reduces its practicality for large-scale graphs and has restricted the broader integration of PH in graph representation learning.

With this work, we take a significant step forward in addressing the challenges of topological graph representation learning and introduce *Topological Evolution Rate (TopER)*. This novel approach refines the Persistent Homology process to efficiently capture graph substructures, thereby mitigating the significant computational demands of calculating complex topological features. As graph representation learning aligns naturally with Topological Data Analysis, *TopER* excels in graph clustering and classification tasks where it achieves the best rank in experiments. Furthermore, simplifying graph data into a low dimensional space, *TopER* creates intuitive visualizations that reveal clusters, outliers, and other essential topological features, as demonstrated in Figure 1. As a result, *TopER* merges interpretability with efficiency in graph representation learning, providing an ideal balance that can scale to large graphs.

To our knowledge, *TopER* is the first topology-based graph representation learning method that can create low-dimensional, efficient, and scalable graph representations.

Our contributions can be summarized as follows:

- **New Graph Embedding Method:** We introduce *TopER*, a compact and computationally feasible graph representation designed to capture the evolution of graph substructures.

- **Interpretable Visualizations:** *TopER* generates interpretable, low-dimensional embeddings, enabling clear visualization of clusters and outliers. It excels in providing insights within individual graph datasets and across multiple datasets, facilitating comparative analysis.

- **Enhanced Use of Persistent Homology:** *TopER* optimizes the filtration process in PH, offering computational efficiency and compact outputs, while boosting the performance of traditional PH methods.

- **Theoretical Stability:** We establish theoretical stability guarantees for *TopER*, ensuring that the embeddings are reliable and robust across various filtration functions.

- **Competitive Performance:** Our extensive experiments on benchmark datasets demonstrate that *TopER* delivers consistently competitive or superior performance in clustering and classification tasks compared to state-of-the-art models.

## 2 RELATED WORK

**Graph Representation Learning.** Graph representation learning is a dynamic subfield of machine learning, focusing on transforming graph data into efficient, low-dimensional vector representations that encapsulate essential features of the data (Hamilton, 2020; Gao et al., 2019). These representations facilitate a deeper analytical understanding of graphs, which is critical for various applications such as molecular graph property prediction (Dong et al., 2019).

**Graph Neural Networks.** GNNs have revolutionized the analysis of graph data, drawing parallels with the success of Convolutional Neural Networks in image processing (Errica et al., 2020). GNNs utilize spectral and spatial approaches to graph convolutions based on the graph Laplacian and direct graph convolutions, respectively (Bruna et al., 2014; Defferrard et al., 2016; Kipf and Welling, 2017). Despite their success, GNNs often suffer from issues like over-smoothing and lack transparency, making them less ideal for applications requiring interpretability (Günnemann, 2022).

**TDA in Graph Representation Learning.** TDA provides a robust and computationally efficient framework to address the interpretability and over-smoothing issues present in GNNs (Aktas et al., 2019). Persistent Homology, a key technique in TDA, has been applied successfully to graph data, demonstrating potential to match or even exceed the performance of traditional methods in classification and clustering tasks (Hensel et al., 2021; Demir et al., 2022; Hiraoka et al., 2024; Immonen et al., 2024; Chen et al., 2024a; Loiseaux et al., 2024). However, the computational intensity of PH limits its scalability (Hofer et al., 2019; Zhao et al., 2020; Akcora et al., 2022).

**Graph Embeddings and Visualization.** Graph embedding techniques, including spectral methods, random walk-based approaches, and deep learning-based models, transform graph data into vector representations to support tasks like visualization and machine learning (Cai et al., 2018; Goyal and Ferrara, 2018; Xu, 2021). Approaches such as Laplacian Eigenmaps and DeepWalk have been particularly effective in revealing clusters within graphs (Belkin and Niyogi, 2001; Perozzi et al., 2014). However, these methods are predominantly applied to visualize a *single graph* in node classification tasks, focusing on cluster identification (Wang et al., 2016; Mavromatis and Karypis, 2020; Tsitsulin et al., 2023). Furthermore, they often overlook domain-specific information, which can limit their effectiveness in more specialized applications (Jin and Zafarani, 2020).

*TopER* addresses these challenges by combining the interpretative benefits of TDA with the analytical strength of modern graph machine learning. Distinct from current approaches, TopER employs a simplified filtration process to create embeddings that are both interpretable and computationally efficient. By extending the filtration to multiple functions, TopER stands out as one of the first methods to offer effective and interpretable visualizations of graph datasets, while also achieving superior performance in clustering and classification tasks.

## 3 BACKGROUND

TDA has emerged as a promising approach in graph representation learning (Aktas et al., 2019). Among its various techniques, persistence homology (PH) is notable for its robust capacity to quantify and monitor topological features across different scales.

### 3.1 PERSISTENT HOMOLOGY FOR GRAPHS

Persistent Homology applies algebraic topology to reveal hidden shape patterns within data, capturing these insights by tracking the evolution of topological features like components, loops, and cavities at varying resolutions (Coskunuzer and Akçora, 2024). The PH process involves constructing filtrations, obtaining persistence diagrams, and vectorizations. Our model, however, primarily utilizes the filtration step, reformulating the evolution information in a novel manner.

In the crucial filtration step, PH decomposes a graph $\mathcal{G}$ into a nested sequence of subgraphs $\mathcal{G}_1 \subseteq \mathcal{G}_2 \subseteq \ldots \subseteq \mathcal{G}_n = \mathcal{G}$. For each $\mathcal{G}_i$, an abstract simplicial complex $\widehat{\mathcal{G}}_i$ is defined, forming a filtration of simplicial complexes. Clique complexes are typical choices, where each $(k+1)$-complete subgraph in $\mathcal{G}$ corresponds to a $k$-simplex (Aktas et al., 2019). To obtain effective filtrations, utilizing relevant filtration functions is essential.

**Filtration Functions.** Filtration functions are essential in PH. For a given graph $\mathcal{G} = (\mathcal{V}, \mathcal{E})$, a common approach is to define a node filtration function $f : \mathcal{V} \to \mathbb{R}$, which establishes a hierarchy among the nodes. By selecting a monotone increasing set of thresholds $\mathcal{I} = \{\epsilon_i\}_{i=1}^n$, this method generates subgraphs $\mathcal{G}_i = (\mathcal{V}_i, \mathcal{E}_i)$ where $\mathcal{V}_i = \{v \in \mathcal{V} \mid f(v) \leq \epsilon_i\}$ and $\mathcal{E}_i$ is the set of edges in $\mathcal{E}$ with endpoints in $\mathcal{V}_i$. This is called a sublevel filtration induced by $f$ (See Figure 2). Similarly, superlevel filtrations can be constructed by defining $\mathcal{V}_i = \{v \in \mathcal{V} \mid f(v) \geq \epsilon_i\}$ for decreasing thresholds (Aktas et al., 2019).

Similarly, one can use edge filtration functions $g : \mathbb{E} \to \mathbb{R}$ to define such a filtration. Similarly, by defining $\mathcal{E}_i = \{e_{jk} \in \mathcal{E} \mid g(e_{jk}) \leq \epsilon_i\}$, and $\mathcal{V}_i$ as the all endpoints of $\mathcal{E}_i$, one can define a nested sequence $\{\mathcal{G}_i\}_{i=1}^n$. Especially, for weighted graphs, this method is highly preferable as weights naturally defines an edge filtration function. The common node filtration functions are degree, betweenness, centrality, heat kernel signatures, and node functions coming from the domain of the datasets (e.g., atomic number for molecular graphs). Common edge filtration functions are Ollivier and Forman Ricci curvatures, and edge weights (e.g. transaction amounts for financial networks).

We developed a new filtration function, *Popularity*, in addition to existing approaches, to enhance the topological representation of graph-structured data in our studies. *Popularity* is similar to the degree function (Newman, 2003), but also considers the average degree of neighboring nodes of a given node $v$. This function enriches node filtration by incorporating broader neighborhood information, potentially revealing deeper insights into graph structure. Mathematically, for each node $v$ in the graph, we define the popularity function as: $\mathcal{P}(v) = \deg(v) + \frac{\sum_{u \in \mathcal{N}(v)} \deg(u)}{|\mathcal{N}(v)|}$, where $\deg(v)$ is the degree of node $v$ and $\mathcal{N}(v)$ denotes the set of nodes adjacent to $v$. This function

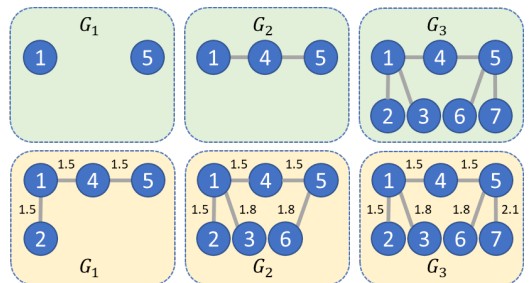

Figure 2: **Filtration.** For $\mathcal{G} = \mathcal{G}_3$ in both examples, the top figure illustrates superlevel filtration with node degree function for thresholds $\{1, 2, 3\}$. Similarly, the bottom figure illustrates sublevel filtration for edge weights with thresholds $\{1.5, 1.8, 2.1\}$.

incorporates 2-neighborhood information by giving more weight to high-degree (popular) neighbors. *Popularity* can be considered an improved version of the degree function, as it also accounts for the popularity of a node's neighbors. The intuition is that if degree represents the number of friends, popularity accounts for the *popular* friends (node neighbors with high degree) with greater weight. Detailed descriptions of filtration functions are given in Appendix D.

## 4 TopER: Topological Evolution Rate

The motivation for this paper is twofold. First, we aim to simplify and achieve computationally feasible outputs in graph representation learning by leveraging the filtration argument in Persistent Homology (PH). Second, we aim to create an effective low-dimensional embedding for graph datasets, bridging the substantial gap in visualization capabilities highlighted earlier.

TopER innovatively extends the concept of filtration used in PH to track graph substructures more efficiently. This reformulation reduces the computational overhead typically required for detailed topological feature extraction. Unlike traditional Persistent Homology, which extracts costly topological features, TopER summarizes the filtration process through two key parameters derived via regression: *filtration sequences* and *evolution*.

**Filtration sequences.** We first decompose a graph $\mathcal{G}$ into a nested sequence of subgraphs (filtration graphs) $\mathcal{G}_1 \subseteq \mathcal{G}_2 \ldots \subseteq \mathcal{G}_n = \mathcal{G}$ by using a filtration function, such as node degree or closeness. Let $\mathcal{G}_i \subset \mathcal{G}$, $\mathcal{V}_i$ represent nodes in $\mathcal{G}_i$ and $\mathcal{E}_i$ represent the edges. Next, we compute $x_i = |\mathcal{V}_i|$ as the count of nodes, and $y_i = |\mathcal{E}_i|$ as the count of edges. Then, for each filtration graph $\mathcal{G}_i$, we obtain the pair $(x_i, y_i) \in \mathbb{R}^2$, which creates two monotone sequences, $x_1 \leq x_2 \leq \cdots \leq x_n$ and $y_1 \leq y_2 \leq \cdots \leq y_n$. Hence, TopER yields two ordered sets $\mathcal{X}, \mathcal{Y}$ describing the evolution of the filtration

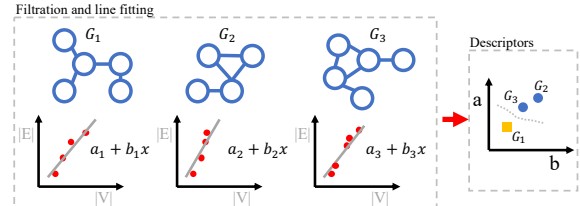

Figure 3: *TopER steps.* The filtration process on three different graphs using node or edge filtration. The graphs undergo filtration, and for each graph, a best-fit line is determined through the filtration data. The coefficients of these best-fit lines are then used as descriptors for the graphs.

graphs $\mathcal{G}_1 \subseteq \ldots \subseteq \mathcal{G}_n = \mathcal{G}$, $\mathcal{X} = (x_1, x_2, \ldots, x_n)$ and $\mathcal{Y} = (y_1, y_2, \ldots, y_n)$. Here, $n$ corresponds to number of thresholds $\{\epsilon_i\}_{i=1}^n$ used in the filtration step.

---

**Algorithm 1** TopER: Topological Evolution Rate

---
1: **Input:** Graph $\mathcal{G}$, Filtration function $f : \mathcal{V} \to \mathbb{R}$, Threshold set $\mathcal{I} = \{\epsilon_i\}_{i=0}^n$
2: **Output:** TopER vector $\mathcal{T}(\mathcal{G}, f, \mathcal{I})$
3:  Initialize lists $\mathcal{X} = [], \mathcal{Y} = []$
4: **for** $i = 1$ to $n$ **do**
5:     $\mathcal{G}_i \leftarrow$ Induced subgraph of $\mathcal{G}$ where $\mathcal{V}_i \subseteq f^{-1}([\epsilon_0, \epsilon_i])$
6:     $x_i \leftarrow |\mathcal{V}_i|$
7:     $y_i \leftarrow |\mathcal{E}_i|$
8:     Append $x_i$ to $\mathcal{X}$
9:     Append $y_i$ to $\mathcal{Y}$
10: **end for**
11: Fit a line $\mathcal{L}(x) = a + bx$ to pairs $(x_i, y_i)$ from lists $\mathcal{X}$ and $\mathcal{Y}$ using least squares
12: Extract coefficients $a$ and $b$
13: **Return** $(a, b)$ as the TopER vector $\mathcal{T}(\mathcal{G}, f, \mathcal{I})$

---

Consider the top row of Figure 2 where we have three filtration graphs (i.e., $n$=3); we have $\mathcal{X} = (2, 3, 7)$ for node counts and $\mathcal{Y} = (0, 2, 6)$ for edge counts.

**Evolution.** In the next step, PH would typically compute topological features on each filtration and create a persistence diagram to summarize the features. Not only it is costly, but the approach would require efforts to vectorize the persistence diagrams. We circumvent this computationally costly step and analyze how the number of edges $\{y_i\}$ relates to the number of nodes $\{x_i\}$ throughout the filtration sequence. We use line fitting to characterize this relationship as follows.

Simple linear regression, often applied through the least squares method (James et al., 2013), is a standard approach in regression analysis for fitting a linear equation to a set of data points $\{(x_i, y_i)\} \subset \mathbb{R}^2$. This method calculates the line $\mathcal{L}(x) = a + bx$ that best fits the data by minimizing the loss function $E = \sum_{i=1}^N [\mathcal{L}(x_i) - y_i]^2$. The coefficients $(a, b)$, obtained from this regression on the filtration sequences, effectively describe the graph's structure during filtration (see the descriptor step in Figure 3). Algorithm 1 unifies the filtration and evolution steps.

With evolution on filtration sequences, we define the topological evolution rate of a graph as follows:

**Definition 1** (Topological Evolution Rate (TopER)). *Let $f : \mathcal{V} \to \mathbb{R}$ be a filtration function on graph $\mathcal{G}$ and $\mathcal{I} = \{\epsilon_i\}_{i=1}^n$ be the threshold set. Let $\mathcal{G}_i = (\mathcal{V}_i, \mathcal{E}_i)$ be the induced filtration. Let $x_i = |\mathcal{V}_i|$ and $y_i = |\mathcal{E}_i|$. Let $\mathcal{L}(x) = a + bx$ be the best fitting line to $\{(x_i, y_i)\}_{i=1}^n$. Then, we define theTopER vector of $\mathcal{G}$ with respect to $f$ as $\text{TE}(\mathcal{G}, f, \mathcal{I}) = (a, b)$. We call $a$ the pivot and $b$ the growth of $\mathcal{G}$.*

In practice, line fitting may not always be a linear choice and requires multiple controls, which we cannot report here due to space limitations. We discuss these choices and linear/polynomial fit options in Appendix B. We also show visual samples of graph evolution rates in Appendix Figure 6. Furthermore, we define and visualize the most fundamental evolution patterns in Appendix Figure 7.

On another note, while PH tracks the evolution of topological changes in the clique complexes of subgraphs $\{\mathcal{G}_i\}$, TopER captures the evolution within $\{\mathcal{G}_i\}$ by monitoring the distributions of nodes and edges as well as connectivity changes in these subgraphs, guided by the hierarchy imposed by the filtration function.

## 4.1 COMPUTATIONAL COMPLEXITY

The primary computational steps in TopER include constructing filtration graphs and performing regression on node and edge counts, which incur the following costs.

Analyzing each node and edge across $n$ filtration thresholds typically requires $O(n \times (|\mathcal{V}| + |\mathcal{E}|))$ operations, where $\mathcal{V}$ and $\mathcal{E}$ denote the numbers of vertices and edges, respectively. The regression step involves fitting a line to the pairs $(x_i, y_i)$ using the least squares method. The complexity of calculating the necessary sums for this regression is $O(n)$, and solving for the regression coefficients (slope and intercept) from these sums involves a constant amount of additional computation.

Thus, the overall complexity of TopER predominantly hinges on the graph filtration process, summing up to $O(n \times (|\mathcal{V}| + |\mathcal{E}|))$ where $|\mathcal{V}| \gg n$. As we will show in the next section, the runtime costs of TopER are notably low, demonstrating its practicality and efficiency for large-scale applications.

## 4.2 STABILITY RESULTS

This section states our theorems on the stability of TopER. In the following, $\mathcal{W}_p(.,.)$ represents $p$-Wasserstein distance, and $\mathrm{PD}_k(\mathcal{X}, f)$ represents $k^{th}$ persistence diagram of $\mathcal{X}$ with sublevel filtration with respect to $f$. Similarly, $\|.\|_p$ represents $L^p$-norm and $d_p(.,.)$ represents $l_p$-distance in $\mathbb{R}^m$. We fix a threshold set $\mathcal{I} = \{\epsilon_i\}_{i=1}^n$ for both functions to keep the exposition simple. Further, to keep the setting general, we use the pairs $\{(\beta_0(\epsilon_i), \beta_1(\epsilon_i))\}_{i=1}^n$ in $\mathbb{R}^2$ to fit the least squares line $y = a + bx$ defining $\mathrm{TE}(\mathcal{X}) = (a, b)$.

**Theorem 1.** *Let $\mathcal{X}$ be a compact metric space, and $f, g : \mathcal{X} \to \mathbb{R}$ be two filtration functions. Then, for some $C > 0$,*

$$\|\mathrm{TE}_f(\mathcal{X}) - \mathrm{TE}_g(\mathcal{X})\|_1 \leq \mathrm{C} \cdot \mathcal{W}_1(\mathrm{PD}_k(\mathcal{X}, f), \mathrm{PD}_k(\mathcal{X}, g)).$$

By combining the above result with the stability result for sublevel filtrations, we obtain the stability with respect to filtration functions as follows.

**Corollary 1.** *Let $\mathcal{X}$ be a compact metric space, and $f, g : \mathcal{X} \to \mathbb{R}$ be two filtration functions. Then, for some $C > 0$,*

$$\|\mathrm{TE}_f(\mathcal{X}) - \mathrm{TE}_g(\mathcal{X})\|_1 \leq \mathrm{C} \cdot \|f - g\|_1$$

By adapting the above results to the graph setting, when two metric graphs $\mathcal{G}_1, \mathcal{G}_2$ are close in Gromov-Hausdorff sense, one can obtain a similar stability result for the filtrations of $\mathcal{G}_i$ induced by the same filtration function. Due to space limitations, details and the proof of the theorem are given in Appendix C.

## 5 EXPERIMENTS

We evaluate the performance of TopER in classification, clustering and visualization. Our Python implementation is available at https://anonymous.4open.science/r/TopER-AA38.

## 5.1 EXPERIMENTAL SETUP

**Datasets.** We conduct experiments on nine benchmark datasets for graph classification. These are (i) the molecule graphs of BZR, and COX2 (Mahé and Vert, 2009); (ii) the biological graphs of MUTAG and PROTEINS (Kriege et al., 2012); and (iii) the social graphs of IMDB-Binary (IMDB-B), IMDB-Multi (IMDB-M), REDDIT-Binary (REDDIT-B), and REDDIT-Multi-5K (REDDIT-5K) (Yanardag and Vishwanathan, 2015). Finally, the OGBG-MOLHIV is a large molecular property prediction dataset, part of open graph benchmark (OGB) datasets (Hu et al., 2020). Details are given in Table 1.

**Hardware.** We ran experiments on a single machine with 12th Generation Intel Core i7-1270P vPro Processor (E-cores up to 3.50 GHz, P-cores up to 4.80 GHz), and 32Gb of RAM (LPDDR5-6400MHz).

**Model Setup and Metrics.** We employ a rigorous experimental setup to ensure a fair comparison and the selection of the best graph classification model. We begin by applying BatchNormalization to the input features to maintain consistent scaling. We employ a 90/10 train-test split, adopt the Stratifiedk-Fold strategy, and present the average accuracy from ten-fold cross-validation across all our models. We employ accuracy as the evaluation metric, a widely utilized performance measure within graph classification tasks (Errica et al., 2020).

Table 1: Characteristics of the benchmark graph classification datasets.

| Datasets | #Graphs | $|\mathcal{V}|$ | $|\mathcal{E}|$ | Classes |
|---|---|---|---|---|
| BZR | 405 | 35.75 | 38.36 | 2 |
| COX2 | 467 | 41.22 | 43.45 | 2 |
| MUTAG | 188 | 17.93 | 19.79 | 2 |
| PROTEINS | 1113 | 39.06 | 72.82 | 2 |
| IMDB-B | 1000 | 19.77 | 96.53 | 2 |
| IMDB-M | 1500 | 13.00 | 65.94 | 3 |
| REDDIT-B | 2000 | 429.63 | 497.75 | 2 |
| REDDIT-5K | 4999 | 508.52 | 594.87 | 5 |
| OGBG-MOLHIV | 41127 | 243.4 | 2266.1 | 2 |

**Filtration functions.** In TopER, we use both node and edge filtrations during filtration (Definition 1). Alongside with popularity, we apply degree, closeness, and degree centrality (Evans and Chen,

Table 2: **Graph Classification.** Accuracy results on eight benchmark datasets using 10-fold CV. Baseline results are sourced from the corresponding papers. The best performance is highlighted in **bold blue**, while the second-best performance is underlined. The final column presents the average deviation of each model's performance from the best result across all datasets.

| Model | BZR | COX2 | MUTAG | PROTEINS | IMDB-B | IMDB-M | REDDIT-B | REDDIT-5K | Avg.↓ |
|---|---|---|---|---|---|---|---|---|---|
| P-WL-C (Rieck et al., 2019) | – | – | $90.51_{\pm1.34}$ | $75.27_{\pm0.38}$ | – | – | – | – | 2.08 |
| 1-GIN (GFL) (Hofer et al., 2020) | – | – | – | $74.10_{\pm3.40}$ | $74.50_{\pm4.60}$ | $49.70_{\pm2.90}$ | $90.20_{\pm2.8}$ | $55.70_{\pm2.90}$ | 2.09 |
| 6 GNNs (Errica et al., 2020) | – | – | $80.42_{\pm2.07}$ | $\underline{75.80}_{\pm3.70}$ | $71.20_{\pm3.90}$ | $49.10_{\pm3.50}$ | $89.90_{\pm1.90}$ | $56.10_{\pm1.60}$ | 4.13 |
| DMP (Bodnar et al., 2021) | – | – | $84.00_{\pm8.60}$ | $75.30_{\pm3.30}$ | $73.80_{\pm4.50}$ | $50.90_{\pm2.50}$ | $86.20_{\pm6.80}$ | $51.90_{\pm2.10}$ | 4.20 |
| FC-V (O'Bray et al., 2021) | $85.61_{\pm0.59}$ | $81.01_{\pm0.88}$ | $87.31_{\pm0.66}$ | $74.54_{\pm0.48}$ | $73.84_{\pm0.36}$ | $46.80_{\pm0.37}$ | $89.41_{\pm0.24}$ | $52.36_{\pm0.37}$ | 4.01 |
| SubMix (Yoo et al., 2022) | $86.34_{\pm2.00}$ | $\underline{84.68}_{\pm3.70}$ | $80.99_{\pm0.60}$ | $67.80_{\pm2.00}$ | $70.30_{\pm1.40}$ | $46.47_{\pm2.50}$ | – | – | 6.15 |
| G-Mix (Han et al., 2022) | $84.15_{\pm2.30}$ | $83.83_{\pm2.10}$ | $81.96_{\pm0.60}$ | $66.28_{\pm1.10}$ | $69.40_{\pm1.10}$ | $46.40_{\pm2.70}$ | – | – | 6.91 |
| RGCL (Li et al., 2022) | $84.54_{\pm1.67}$ | $79.31_{\pm0.68}$ | $87.66_{\pm1.01}$ | $75.03_{\pm0.43}$ | $71.85_{\pm0.84}$ | $49.31_{\pm0.42}$ | $90.34_{\pm0.58}$ | $56.38_{\pm0.40}$ | 3.56 |
| AutoGCL (Yin et al., 2022) | $86.27_{\pm0.71}$ | $79.31_{\pm0.70}$ | $88.64_{\pm1.08}$ | $\underline{75.80}_{\pm0.36}$ | $72.32_{\pm0.93}$ | $50.60_{\pm0.80}$ | $88.58_{\pm1.49}$ | $\mathbf{56.75}_{\pm0.18}$ | 3.08 |
| FF-GCN (Paliotta et al., 2023) | $\underline{89.00}_{\pm5.00}$ | $78.00_{\pm8.00}$ | $71.00_{\pm4.00}$ | $62.00_{\pm1.00}$ | $63.00_{\pm8.00}$ | – | – | – | 11.53 |
| WWLS (Fang et al., 2023) | $88.02_{\pm0.61}$ | $81.58_{\pm0.91}$ | $88.30_{\pm1.23}$ | $75.35_{\pm0.74}$ | $\mathbf{75.08}_{\pm0.31}$ | $\underline{51.61}_{\pm0.62}$ | – | – | 2.26 |
| EPIC (Heo et al., 2024) | $88.78_{\pm2.30}$ | $\mathbf{85.53}_{\pm1.60}$ | $82.44_{\pm0.70}$ | $69.06_{\pm1.00}$ | $71.70_{\pm1.00}$ | $47.93_{\pm1.30}$ | – | – | 4.67 |
| EMP (Chen et al., 2024a) | – | – | $88.79_{\pm0.63}$ | $72.78_{\pm0.54}$ | $74.44_{\pm0.45}$ | $48.01_{\pm0.42}$ | $\underline{91.03}_{\pm0.22}$ | $54.41_{\pm0.32}$ | 2.97 |
| MP-HSM (Loiseaux et al., 2024) | – | $77.10_{\pm3.00}$ | $85.60_{\pm5.30}$ | $74.60_{\pm2.10}$ | $\underline{74.80}_{\pm2.50}$ | $47.90_{\pm3.20}$ | – | – | 4.67 |
| TopoGCL (Chen et al., 2024b) | $87.17_{\pm0.83}$ | $81.45_{\pm0.55}$ | $90.09_{\pm0.93}$ | $\mathbf{77.30}_{\pm0.89}$ | $74.67_{\pm0.32}$ | $\mathbf{52.81}_{\pm0.31}$ | $90.40_{\pm0.53}$ | – | $\underline{1.76}$ |
| PGOT (Qian et al., 2024) | $87.32_{\pm3.90}$ | $82.98_{\pm5.21}$ | $\mathbf{92.63}_{\pm2.58}$ | $73.21_{\pm2.59}$ | $62.90_{\pm3.05}$ | $51.33_{\pm1.76}$ | – | – | 3.85 |
| RePHINE (Immonen et al., 2024) | – | – | – | $71.25_{\pm1.60}$ | $69.40_{\pm3.78}$ | – | – | – | 5.86 |
| **TopER** | $\mathbf{90.13}_{\pm4.14}$ | $82.01_{\pm4.59}$ | $\underline{90.99}_{\pm6.64}$ | $74.58_{\pm3.92}$ | $73.20_{\pm3.43}$ | $50.00_{\pm4.02}$ | $\mathbf{92.70}_{\pm2.38}$ | $\underline{56.51}_{\pm2.22}$ | $\mathbf{1.60}$ |

2022) as node filtration functions and Forman- and Ollivier-Ricci functions (Lin et al., 2011) as edge filtration functions. We also use atomic weight as a node function for molecular and biological datasets (BZR, COX2, and MUTAG), and node attributes (PROTEINS). We utilized the t-test to assess the statistical significance of each function and applied the Lasso method for regularization, setting the cross-validation parameter to $cv = 10$. Functions were retained in the model only if they achieved p-values less than 0.05 in the t-test and had non-zero coefficients in the Lasso model (James et al., 2023). This approach ensures that the selected filtration functions contribute statistically significant and regularized features to the model. Incorporating additional filtration functions can enhance TopER's ability to analyze graphs from diverse perspectives. However, as we will next illustrate in Table 6, TopER demonstrates strong performance even in its most basic form using the simple and scalable node degree function. This balance of performance and simplicity suits our scalability philosophy; we avoid complex and costly schemes for learning dataset-specific activation functions and homogenize the filtration step in all datasets.

**Classifier.** We utilize a Multilayer Perceptron (MLP) in our graph classification task. The hyperparameters are detailed in Appendix A.1.

**Baselines.** We compare our method with 19 state-of-the-art and recent models in graph classification, including variants of graph neural networks: six GNNs including GCN, DGCNN, Diffpool, ECC, GIN, GraphSAGE which are compared in (Errica et al., 2020) (best results of these six GNNs are given in the *6 GNNs row*), FF-GCN (Paliotta et al., 2023); topological methods: DMP (Bodnar et al., 2021), FC-V (O'Bray et al., 2021), WWLS (Fang et al., 2023), MP-HSM (Loiseaux et al., 2024) and EMP (Chen et al., 2024a); GNNs enhanced with data augmentation methods: SubMix (Yoo et al., 2022), G-Mix (Han et al., 2022), and EPIC (Heo et al., 2024); GNNs enhanced with contrastive learning methods: RGCL (Li et al., 2022), AutoGCL (Yin et al., 2022), TopoGCL (Chen et al., 2024b) and prototype-based methods: PGOT (Qian et al., 2024).

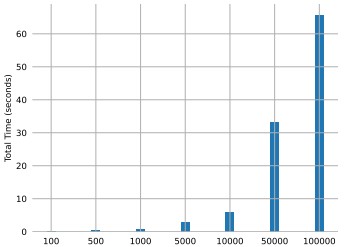

Figure 4: TopER run time for synthetic power law graphs (Holme and Kim, 2002) with node degree filtration. The mean node degree is 30, and 100 filtration steps are used.

**Runtime.** TopER is highly scalable and can be applied to a 100K node graph in ~2 minutes (see Figure 4). Our small network experiments took about two days in a shared resource setting, whereas the OGBG-MOLHIV experiments took 16.5 hours. One of the most demanding datasets, REDDIT-5K, requires 34.6 hours to calculate all node and edge functions. The runtime of our methods is dominated by the computation of node functions such as closeness and Riccis. Using approximate values for centrality metrics instead could greatly decrease computation time (Brandes and Pich, 2007). Since this is not our current focus, we leave it as future work.

## 5.2 GRAPH CLASSIFICATION RESULTS

Table 2 shows the accuracy results for the given models. We use the reported results in the corresponding references for each model. "−" entries in the table mean the reference did not report any result for that dataset. In (Errica et al., 2020), the authors compare the six most common GNNs on the graph classification task (see the GNNs row). The last column summarizes each model's overall performance. We report the average of the differences between each model's performance and the best performance in the column across all datasets. If a model's performance is missing for a dataset, it is excluded from the average computation for the model.

Out of eight datasets, TopER achieves the best results in two and ranks second in two other datasets. For the remaining four datasets, TopER's performance is within 4% of the SOTA results. *For overall performance, TopER outperforms all other models with an average deviation of 1.60% from the best performances.* The closest competitor is TopoGCL, which has an average deviation of 1.76%.

**OGBG-MOLHIV results.** To evaluate our model's performance on large datasets, we compare it with recently published models on OGBG-MOLHIV dataset, as shown in Table 3. The performances of these models are listed in chronological order based on their publication dates, with baseline performances reported from (Choi et al., 2022; Ying et al., 2021) or the respective model's references. In Appendix A.2, we give further details for TopER performance and contribution of each function on this dataset. TopER achieves the second-best result on the MOLHIV dataset, while the top-performing model requires learning a significantly larger model with 119.5 million parameters.

Table 3: AUC results for OGBG-MOLHIV dataset.

| Model | AUC |
|---|---|
| GIN-VN (Xu et al., 2018) | $77.80_{\pm 1.82}$ |
| HGK-WL (Togninalli et al., 2019) | $79.05_{\pm 1.30}$ |
| WWL (Borgwardt et al., 2020) | $75.58_{\pm 1.40}$ |
| PNA (Corso et al., 2020) | $79.05_{\pm 1.32}$ |
| DGN (Beaini et al., 2021) | $79.70_{\pm 0.97}$ |
| GraphSNN (Wijesinghe and Wang, 2021) | $79.72_{\pm 1.83}$ |
| GCN-GNorm (Cai et al., 2021) | $78.83_{\pm 1.00}$ |
| Graphormer (Ying et al., 2021) | $\mathbf{80.51}_{\pm 0.53}$ |
| Cy2C-GCN (Choi et al., 2022) | $78.02_{\pm 0.60}$ |
| GAWL (Nikolentzos and Vazirgiannis, 2023) | $78.34_{\pm 0.39}$ |
| LLM-GIN (Zhong et al., 2024) | $79.22_{\pm \text{NA}}$ |
| GMoE-GIN (Wang et al., 2024) | $76.90_{\pm 0.90}$ |
| TopER | $\underline{80.21}_{\pm 0.15}$ |

**TopER vs. PH.** TopER-based models are more accurate than those based on Persistent Homology. We present a comparison across six datasets in Table 4. The PH results are obtained from (Cai, 2021), which extensively examines PH methods in graph classification tasks using four common filtration functions (centrality, degree, fiedler_s, and Ricci), coupled with four popular vectorization techniques. For each dataset, these methods yield 16 outcomes, and we bring in the top-performing combination of filtration and vectorization for each dataset. The results show that TopER outperforms PH in all benchmark datasets. For time comparison between PH and TopER, see Appendix A.5.

Table 4: Accuracy results for TopER vs. Persistent Homology in graph classification tasks.

| | BZR | COX2 | PROTEINS | IMDB-B | IMDB-M | RED-5K |
|---|---|---|---|---|---|---|
| PH | $88.4_{\pm 0.6}$ | $\mathbf{82.0}_{\pm 0.6}$ | $74.0_{\pm 0.4}$ | $69.5_{\pm 0.5}$ | $46.5_{\pm 0.3}$ | $54.1_{\pm 0.1}$ |
| TopER | $\mathbf{90.1}_{\pm 4.1}$ | $\mathbf{82.0}_{\pm 4.6}$ | $\mathbf{74.6}_{\pm 3.9}$ | $\mathbf{73.2}_{\pm 3.4}$ | $\mathbf{50.0}_{\pm 4.0}$ | $\mathbf{56.5}_{\pm 2.2}$ |

## 5.3 GRAPH CLUSTERING RESULTS

We employ cluster quality metrics to assess the embeddings of graphs sourced from all datasets in Table 2. The embeddings are labeled with their respective dataset memberships, and we assume that good embeddings will have graphs of the same dataset clustered together. We evaluate embeddings based on three widely used clustering metrics: Silhouette (SILH), Calinski-Harabasz (CH), and Davies-Bouldin (DB) (Gagolewski et al., 2021). Table 5 compares the clustering performance of

Table 5: **Clustering Performances.** Comparison of Spectral Zoo vs. TopER. The detailed results are given in Appendix A.3.

| Metric | Method | BZR | COX2 | MUTAG | PROT. | IMDB-B | IMDB-M | REDD-B | REDD-5K |
|---|---|---|---|---|---|---|---|---|---|
| Silh ↑ | Spec. Zoo | 0.050 | 0.049 | **0.344** | 0.050 | **0.097** | **-0.024** | 0.108 | -0.121 |
| | TopER | **0.249** | **0.414** | 0.258 | **0.086** | 0.064 | -0.032 | **0.196** | **-0.067** |
| CH ↑ | Spec. Zoo | 3.51 | 6.13 | **120.73** | 38.77 | **85.24** | **30.98** | 269.94 | 119.81 |
| | TopER | **42.58** | **26.00** | 72.52 | **151.64** | 60.52 | 11.77 | **446.12** | **1209.95** |
| DB ↓ | Spec. Zoo | 7.25 | 6.07 | 0.95 | 4.55 | 2.78 | 10.73 | 2.20 | 25.74 |
| | TopER | **1.93** | **2.29** | **0.88** | **1.54** | **2.19** | **6.87** | **1.32** | **2.78** |

TopER and Spectral Zoo (Jin and Zafarani, 2020), which is, to our knowledge, the only model that allows low-dimensional graph embeddings. Detailed results are provided in Appendix A.3. The findings demonstrate that the embeddings generated by TopER outperform those created by Spectral Zoo. This is evident from the superior cluster quality metrics observed for five out of eight datasets in the case of Silhouette and CH, and for all eight datasets in the case of DB.

## 5.4 GRAPH VISUALIZATION

In the case of a single filtration function, TopER creates $2D$ graph embeddings $(a, b)$ that can be visualized with ease (see Figure 1). Traditional dimensionality reduction techniques such as PCA can be used to visualize point cloud data, but accurately depicting graph data has historically been a significant challenge (Giovannangeli et al., 2020). To our knowledge, the only model that allows graph visualization is the GraphZoo (Jin and Zafarani, 2020).

TopER creates highly interpretable graph visualizations. To recall, the pair $(a, b)$ represents the coefficients of the best-fitting function $L(x) = a + bx$, where $a$ is the pivot (y-intercept) and $b$ is growth (the slope). Specifically, the pivot $a$ reflects graph connectivity, while $b$ reflects the growth rate of edges/nodes for the filtration function. In particular, a higher value of $a$ corresponds to a more interconnected graph. As we demonstrate in Figure 7, graph connectivity and community structure can be analyzed using three types of pivot behavior. In the following, we illustrate how these quantities can be employed to interpret our two-dimensional representations of the graph datasets.

In Figure 1b of the MUTAG dataset, class B has a higher growth rate and smaller pivot than the red class. This shows that the class is growing faster than class A with respect to the closeness function in the MUTAG dataset, i.e., the graph has a low diameter. Similarly, in contrast, in the PROTEINS dataset (Figure 5a), the growth rates are similar for both classes ($\sim 1.5 - 1.7$), but the pivot (initial graph size) is smaller in class A. This implies that class A has fewer edges in relation to the number of nodes. Such patterns, as described in Appendix B.4, can reveal key insights into graph topology. In a similar vein, TopER visualizations can be used for anomaly detection. For example, in Figure 1a, an outlier

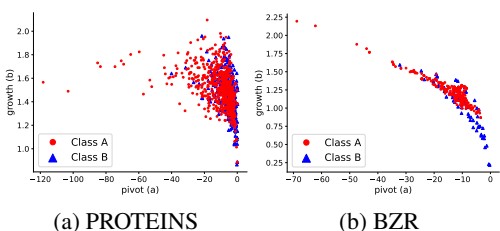

(a) PROTEINS      (b) BZR

Figure 5: TopER visualizations of the PROTEINS dataset with Ollivier-Ricci edge filtration, and the BZR dataset with degree centrality node filtration. Each point corresponds to an individual graph.

PROTEINS graph alone has a positive pivot in the figure and appears as the rightmost data point.

More importantly, **TopER homogenizes graph embeddings, allowing us to compare graphs across datasets**, which may open new paths in training graph foundation models. For example, Figure 1a visualizes graphs of three datasets on the same panel where we see that Mutag and COX2 differ in their pivot only. The similarity is not surprising; MUTAG and COX2 are datasets of molecular graphs where nodes are atoms and edges are chemical bonds. As the molecules in both datasets have similar types of atoms and bond configurations (e.g., ring structures), TopER captures these similarities, leading to similar embeddings.

## 5.5 ABLATION STUDY

In an ablation study, we evaluate the individual performance of each function as well as their combined effect on classification. As shown in Table 6, the common filtration functions we employ from TDA exhibit strong individual performance. Moreover, when combined, they synergistically enhance overall performance. This is not surprising, as different filtration functions—such as atomic weight or Ricci curvature—generate distinct hierarchies and node-edge distributions, resulting in diverse connectivity patterns throughout the filtration sequence. This diversity is analogous to viewing an object from multiple angles. Hence, integrating these complementary perspectives improves performance by offering a richer and more varied representation of the graph structure, allowing the model to capture more intricate features.

Table 6: **Ablation Study.** Individual and altogether performances of filtration functions with TopER.

| Datasets | Degree-cent. | Popularity | Closeness | Degree | F. Ricci | O. Ricci | Atom weight | TopER |
|---|---|---|---|---|---|---|---|---|
| BZR | $82.22_{\pm2.13}$ | $82.20_{\pm3.42}$ | $81.48_{\pm1.99}$ | $\underline{82.73}_{\pm2.12}$ | $80.75_{\pm1.73}$ | $80.99_{\pm1.48}$ | $82.23_{\pm2.12}$ | $\mathbf{90.13}_{\pm4.14}$ |
| COX2 | $\underline{75.38}_{\pm3.96}$ | $69.21_{\pm8.19}$ | $67.90_{\pm7.96}$ | $73.88_{\pm5.02}$ | $70.46_{\pm7.28}$ | $73.03_{\pm4.21}$ | $69.82_{\pm8.27}$ | $\mathbf{82.01}_{\pm4.59}$ |
| MUTAG | $76.61_{\pm7.87}$ | $77.66_{\pm6.12}$ | $80.88_{\pm4.79}$ | $74.97_{\pm6.40}$ | $80.85_{\pm9.25}$ | $\underline{82.46}_{\pm7.84}$ | $73.45_{\pm8.01}$ | $\mathbf{90.99}_{\pm6.64}$ |
| PROTEINS | $67.66_{\pm3.16}$ | $70.71_{\pm4.41}$ | $69.01_{\pm4.24}$ | $69.01_{\pm3.48}$ | $72.96_{\pm3.47}$ | $71.25_{\pm2.66}$ | $\underline{73.59}_{\pm3.33}$ | $\mathbf{74.58}_{\pm3.92}$ |
| IMDB-B | $73.00_{\pm4.49}$ | $71.90_{\pm3.48}$ | $72.60_{\pm4.20}$ | $\underline{73.10}_{\pm4.18}$ | $69.80_{\pm2.44}$ | $66.40_{\pm3.35}$ | - | $\mathbf{73.20}_{\pm3.43}$ |
| IMDB-M | $\underline{48.47}_{\pm3.90}$ | $47.87_{\pm3.07}$ | $48.33_{\pm3.49}$ | $47.93_{\pm2.88}$ | $48.13_{\pm4.11}$ | $43.60_{\pm3.17}$ | - | $\mathbf{50.00}_{\pm4.02}$ |
| REDDIT-B | $76.70_{\pm3.69}$ | $79.35_{\pm3.46}$ | $78.10_{\pm3.23}$ | $\underline{79.55}_{\pm2.20}$ | $72.35_{\pm2.91}$ | $68.20_{\pm2.28}$ | - | $\mathbf{92.70}_{\pm2.38}$ |
| REDDIT-5K | $42.85_{\pm1.74}$ | $\underline{50.87}_{\pm2.63}$ | $50.03_{\pm1.49}$ | $47.01_{\pm1.89}$ | $50.27_{\pm1.92}$ | $45.81_{\pm2.08}$ | - | $\mathbf{56.51}_{\pm2.22}$ |

We further present two ablation studies. In Appendix A.4, we present the effect of number of thresholds on the performance of TopER. In Appendix A.6, the effect of the number of filtration functions used on TopER's performance.

## 6 CONCLUSION

We have introduced a novel graph embedding method, *TopER*, leveraging Persistent Homology from Topological Data Analysis. *TopER* demonstrates strong performance in graph classification tasks, rivaling SOTA models. Furthermore, it naturally generates effective 2D visualizations of graph datasets, facilitating the identification of clusters and outliers. For future research, one promising direction is to extend *TopER* to temporal graph learning tasks, enabling the capture of dynamic graph trajectories that reflect evolving user behaviors over time. Another avenue worth exploring is the integration of *TopER* embeddings into graph foundation models, where the homogenization of graph structures could enhance the learning of transferable representations across different domains.

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

# A    FURTHER EXPERIMENTAL DETAILS

## A.1    HYPERPARAMETERS

Our proposed MLP algorithm is constructed with a single hidden layer. The output layer's activation function is set to log softmax, and the loss function we used is Negative Log Likelihood Loss. The learning rate is chosen between 0.01 and 0.001. Subsequently, we investigate the impact of the number of neurons in the hidden layer, considering values from the set {16, 64, 128}, the optimizer is set to be Adam, and the number of epochs is 500. To prevent large weights and overfitting, we apply L2 regularization coefficients of 1e-3, 1e-4. The activation function for the hidden layer varies between relu, gelu, and elu. Lastly, we consider the cases of adding or not a batch normalization layer to the output of the hidden layer and set dropout values to be 0.0 or 0.5. In Table 7, we provide the details for each dataset. The last column shows the number of TopER features used for each dataset after the feature selection step.

Table 7: Employed hyperparameters for each dataset.

| Dataset | Neurons | Dropout | Batch Norm. | Decay | Learning rate | Activation | TopER Dim. |
|---|---|---|---|---|---|---|---|
| **BZR** | 64 | 0.5 | True | 1e-4 | 0.001 | gelu | 26 |
| **COX2** | 128 | 0 | True | 1e-4 | 0.01 | relu | 26 |
| **MUTAG** | 16 | 0.5 | False | 1e-3 | 0.01 | gelu | 20 |
| **PROTEINS** | 64 | 0.5 | True | 1e-3 | 0.01 | elu | 26 |
| **IMDB-B** | 128 | 0 | False | 1e-3 | 0.001 | relu | 20 |
| **IMDB-M** | 16 | 0 | False | 1e-3 | 0.01 | elu | 20 |
| **REDDIT-B** | 64 | 0.5 | False | 1e-3 | 0.01 | relu | 24 |
| **REDDIT-5K** | 128 | 0 | False | 1e-3 | 0.01 | elu | 14 |

## A.2    OGBG-MOLHIV RESULTS

For OGBG-MOLHIV dataset, we further evaluated the improvements of TopER with addition of new filtration functions. Table 8 provides the performance of each TopER$-i$, where $i$ represents number of filtration functions used in the model, i.e., TopER-$i$ uses $\{(a_1, b_1, \ldots, a_i, b_i)\}$ as graph embedding where $(a_i, b_i)$ is the pivot and growth for function $f_i$. We used XGBoost to rank the importance of filtration functions first, and the functions are added iteratively with this ranking. We fixed maximum tree depth = 3, learning rates = 0.035, subsample ratios = 0.95, the number of estimators = 1000, and the regularization parameter lambda = 45, where the objective function is rank:pairwise, with log loss as the evaluation metric. The seed is set to be 16.

Table 8: Results for OGBG-MOLHIV of each TopER$-i$.

| Method | Added Function | Filtration | Valid. AUC | Test AUC |
|---|---|---|---|---|
| TopER-1 | degree-centrality | sublevel | $72.76_{\pm 0.23}$ | $74.44_{\pm 0.20}$ |
| TopER-2 | atomic weight | sublevel | $71.89_{\pm 0.12}$ | $74.25_{\pm 0.16}$ |
| TopER-3 | Ollivier Ricci | sublevel | $70.11_{\pm 0.28}$ | $76.79_{\pm 0.24}$ |
| TopER-4 | Forman Ricci | superlevel | $71.76_{\pm 0.18}$ | $78.15_{\pm 0.15}$ |
| TopER-5 | degree | superlevel | $71.79_{\pm 0.35}$ | $79.26_{\pm 0.14}$ |
| TopER-6 | popularity | superlevel | $72.27_{\pm 0.29}$ | $79.88_{\pm 0.24}$ |
| TopER-7 | closeness | sublevel | $71.30_{\pm 0.18}$ | $\mathbf{80.21_{\pm 0.15}}$ |

## A.3    CLUSTERING PERFORMANCES

In Table 9, we showcase our clustering performance across eight benchmark graph classification datasets using three widely adopted clustering metrics: Silhouette, Calinski-Harabasz, and Davies-Bouldin. These metrics serve as evaluative measures for assessing the efficacy of clustering algorithms in partitioning datasets into meaningful clusters. They gauge the degree of similarity or dissimilarity within and between clusters, offering insights into the quality of clustering outcomes. For precise definitions of Silhouette, Calinski-Harabasz, and Davies-Bouldin metrics, as well as additional details on clustering measures, refer to Gagolewski et al. (2021).

Table 9: The clustering performances of Spectral Embeddings and TopER with different metrics. Best performances are given in **blue**.

**Silhouette Scores (↑)**

| Method | BZR | COX2 | MUTAG | PROT. | IMDB-B | IMDB-M | REDD-B | REDD-5K |
|---|---|---|---|---|---|---|---|---|
| Spec Zoo | 0.050 | 0.049 | **0.344** | 0.050 | **0.097** | **-0.024** | 0.108 | -0.121 |
| degree | -0.108 | **0.414** | 0.258 | 0.048 | 0.030 | -0.032 | 0.049 | -0.169 |
| popularity | **0.249** | -0.015 | 0.134 | -0.000 | 0.008 | -0.159 | **0.196** | -0.173 |
| closeness | 0.019 | 0.036 | 0.036 | **0.086** | nan | nan | 0.087 | -0.185 |
| degree | 0.084 | 0.030 | 0.017 | 0.065 | 0.056 | -0.075 | 0.034 | **-0.067** |

**Calinski-Harabasz scores (↑)**

| Method | BZR | COX2 | MUTAG | PROT. | IMDB-B | IMDB-M | REDD-B | REDD-5K |
|---|---|---|---|---|---|---|---|---|
| Spec Zoo | 3.51 | 6.13 | **120.73** | 38.77 | **85.24** | **30.98** | 269.94 | 119.81 |
| degree | 0.42 | 1.06 | 11.29 | 130.07 | 60.52 | 3.92 | 97.85 | **1209.95** |
| popularity | 13.85 | **26.00** | 36.13 | 77.22 | 12.89 | 11.77 | **446.12** | 619.37 |
| closeness | **42.58** | 1.02 | 40.04 | 73.51 | 10.17 | 0.30 | 188.10 | 689.27 |
| F.Ricci | 4.92 | 0.48 | 11.82 | **151.64** | 11.68 | 1.03 | 92.14 | 454.34 |

**Davies-Bouldin scores (↓)**

| Method | BZR | COX2 | MUTAG | PROT. | IMDB-B | IMDB-M | REDD-B | REDD-5K |
|---|---|---|---|---|---|---|---|---|
| Spec Zoo | 7.25 | 6.07 | 0.95 | 4.55 | 2.78 | 10.73 | 2.20 | 25.74 |
| degree | 9.84 | **2.29** | **0.88** | 1.95 | 4.92 | 46.46 | 2.32 | 3.27 |
| popularity | 4.16 | 37.87 | 1.62 | 2.11 | 25.25 | **6.87** | **1.32** | 3.46 |
| closeness | **1.93** | 26.44 | 1.41 | 2.25 | 4.99 | 37.51 | 1.95 | **3.09** |
| F.Ricci | 4.19 | 7.20 | 1.27 | **1.54** | **2.19** | 10.35 | 1.83 | 5.41 |

## A.4 NUMBER OF THRESHOLDS

In our experiments, we utilized a large number of thresholds to capture finer-grained information, as the model is computationally efficient and the additional cost of increasing the number of thresholds is minimal. Furthermore, in Table 10, we evaluated the model's performance with fewer thresholds and observed that it remains robust and highly effective even in such scenarios.

Table 10: The accuracy results of TopER with different number of thresholds.

| # Thresholds | PROTEINS | REDDIT-B | REDDIT-5K |
|---|---|---|---|
| 10 | $72.78_{\pm 4.04}$ | $90.55_{\pm 1.96}$ | $55.99_{\pm 1.97}$ |
| 20 | $74.31_{\pm 3.23}$ | $91.20_{\pm 1.66}$ | $55.91_{\pm 2.14}$ |
| 50 | $74.76_{\pm 4.55}$ | $92.05_{\pm 1.96}$ | $55.39_{\pm 2.10}$ |
| 100 | $73.85_{\pm 3.67}$ | $92.85_{\pm 1.18}$ | $55.51_{\pm 2.61}$ |
| 200 | $75.47_{\pm 3.06}$ | $93.15_{\pm 2.10}$ | $56.51_{\pm 2.04}$ |
| 500 | $74.58_{\pm 3.92}$ | $92.70_{\pm 2.38}$ | $56.51_{\pm 3.22}$ |

## A.5 TIME EXPERIMENTS FOR TOPER VS. PH

To compare the time efficiency and performance of TopER and persistent homology (PH), we conducted experiments using the same filtration function, the sublevel degree filtration. For PH, we applied Betti vectorization. Our results, summarized below, show that TopER is significantly faster than PH. Although both methods use the same filtration function, a key distinction lies in their embeddings: TopER generates 2D embeddings, whereas PH produces a vector with dimensionality equal to the number of thresholds in the filtration. Despite the considerable difference in dimensionality, TopER's performance with 2D embeddings remains comparable to that of PH.

## A.6 COMBINING FILTRATION FUNCTIONS

To assess the impact of embedding dimensions, we conducted new experiments evaluating the performance of the TopER model by progressively adding each filtration function step by step.

Table 11: Comparison of TopER-1 and PH in terms of time and accuracy across different datasets.

| Dataset | TopER-1 | | PH | | |
|---|---|---|---|---|---|
| | Time | Accuracy | Time | Accuracy | # Thresholds |
| BZR | 3.13 s | 82.73 ± 2.12 | 5.99 s | 83.70 ± 3.51 | 4 |
| IMDB-B | 16.52 s | 73.10 ± 4.18 | 319.95 s | 71.00 ± 4.07 | 65 |
| REDDIT-B | 11 min 40.60 s | 79.55 ± 2.20 | 152 min 53.37 s | 84.50 ± 2.51 | 501 |

This analysis provides insights into how the inclusion of additional filtration functions influences the model's performance. In Table 12, TopER-n model represents the TopER utilizing n-filtration functions (2n features). In Table 13 we give the corresponding filtration function and filtration type for TopER-n models for each dataset.

Table 12: Performance improvements achieved by integrating filtration functions into the TopER model. Here, TopER-n denotes the TopER model with n filtration functions.

| Dataset | TopER-1 | TopER-2 | TopER-3 | TopER-4 |
|---|---|---|---|---|
| BZR | $82.48_{\pm1.98}$ | $84.70_{\pm2.84}$ | $85.66_{\pm5.00}$ | $86.68_{\pm3.81}$ |
| COX2 | $78.81_{\pm1.94}$ | $79.26_{\pm4.86}$ | $79.04_{\pm7.49}$ | $80.30_{\pm3.91}$ |
| MUTAG | $86.14_{\pm6.38}$ | $88.33_{\pm3.88}$ | $86.75_{\pm4.78}$ | $88.30_{\pm4.63}$ |
| PROTEINS | $74.03_{\pm2.71}$ | $74.67_{\pm2.73}$ | $75.21_{\pm3.39}$ | $75.65_{\pm3.87}$ |
| IMDB-B | $73.00_{\pm4.40}$ | $74.20_{\pm4.26}$ | $74.50_{\pm3.50}$ | $74.70_{\pm3.95}$ |
| IMDB-M | $48.73_{\pm4.33}$ | $49.80_{\pm2.94}$ | $49.73_{\pm4.18}$ | $49.87_{\pm4.00}$ |
| REDDIT-B | $81.95_{\pm2.74}$ | $90.45_{\pm2.55}$ | $91.05_{\pm2.62}$ | $91.50_{\pm2.01}$ |
| REDDIT-5K | $50.21_{\pm1.41}$ | $54.11_{\pm2.43}$ | $56.19_{\pm2.40}$ | $56.33_{\pm2.74}$ |

Table 13: The filtration functions used in TopER-n models in Table 12.

| Dataset | TopER-1 | TopER-2 | TopER-3 | TopER-4 |
|---|---|---|---|---|
| BZR | atomic sub | atomic super | popularity super | degree super |
| COX2 | closeness super | degree cent sub | atomic super | closeness sub |
| MUTAG | O. Ricci sub | F. Ricci super | degree sub | popularity super |
| PROTEINS | atomic sub | closeness sub | O. Ricci super | degree sub |
| IMDB-B | popularity super | popularity sub | degree cent sub | degree cent super |
| IMDB-M | popularity sub | F. Ricci super | closeness super | popularity super |
| REDDIT-B | O. Ricci super | closeness super | degree super | popularity super |
| REDDIT-5K | closeness sub | F. Ricci super | closeness super | popularity super |

# B  MORE ON TOPER

## B.1  REFINING THE POINT SET

While we have described the main steps of TopER in Section 4, due to the repetitions of the points in $\mathcal{A} = \{(x_i, y_i)\} \subset \mathbb{R}^2$, there are some choices to be made before defining the set $\mathcal{A}$ (i.e., $\mathcal{X}$ and $\mathcal{Y}$) to get the best fitting function $L : \mathcal{X} \to \mathcal{Y}$. The main reason is that the set $\{(x_i, y_i)\}_{i=1}^N$ can contain repetitions of $x$-values ($x_i = x_{i+1}$), repetitions of $y$-values ($y_i = y_{i+1}$) or repetitions of both ($(x_i, y_i) = (x_{i+1}, y_{i+1})$) depending on the filtration function, the threshold set $\mathcal{I}$, and the graph $\mathcal{G}$.

For the filtrations induced by *node filtration functions*, the number of edges can not change unless the number of nodes changes, i.e., $x_i = x_{i+1} \Rightarrow y_i = y_{i+1}$. Hence, with this elimination, we still allow keeping $y$-values the same while $x$-values are increasing. This means there can be horizontal jumps in $\mathcal{A}_u$. In this paper, to eliminate all horizontal jumps for filtrations with node functions, we eliminate all repetitions of $y$-values from $\mathcal{A}_u$. AIn particular, we remove all the points with the same $\widehat{y}$-value and add a point with a mean of $x$-values. In other words, if $y_i = y_{i+1} = \cdots = y_{i+k} = \widehat{y}$, we define $\widehat{x} = \text{mean}\{x_i, x_{i+1}, \ldots, x_{i+k}\}$. Then, we replace (k+1) points $\{(x_i, \widehat{y}), (x_{i+1}, \widehat{y}), \ldots, (x_{i+k}, \widehat{y})\}$ with one point $(\widehat{x}, \widehat{y})$ in $\mathcal{A}_u$. This process eliminates all repetitions and horizontal jumps in $\mathcal{A}$, and we define our best-fitting line on this refined set.

## B.2 TopER with Alternative Quantities

While we use the most general quantities for a graph—the count of vertices and edges—in our algorithm, depending on the problem, there might be other induced quantities $(x_i, y_i)$ for a given subgraph $\mathcal{G}_i$ which can give better vectors. To keep the line-fitting approach meaningful in our model, as long as the sequences $\{x_i\}$ and $\{y_i\}$ are monotone like our node-edge counts above, for a given dataset in a domain (e.g., biochemistry, finance), one can use other domain-related quantities induced by substructure $\mathcal{G}_i$ as $(x_i, y_i)$ pair to obtain a TopER vector.

## B.3 Linear or Higher Order Fitting

In our experiments, we observe that linear fitting captures the growth information for node-edge pair $\{(x_i, y_i)\}$ well (See Figure 6), and quadratic fit and linear fit stays very close to each other. However, if one decides to use other quantities as described above and loses the monotonicity of the sequences $\{x_i\}$ and $\{y_i\}$, trying higher order fits (e.g., $y = ax^2 + bx + c$) can be more meaningful. In Table 14, we present the average of the coefficients of quadratic terms when we use quadratic fit for the datasets, i.e. if we fit $y = a + bx + cx^2$ polynomial, we observe that quadratic term $cx^2$ is mostly negligible, and the tends to be a linear fit.

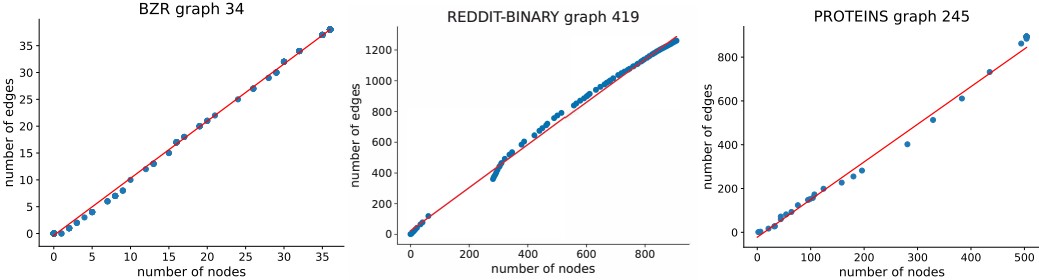

Figure 6: **Linear Fit.** TopER summarizes the growth behavior in the graph induced by filtration with a linear fit.

Table 14: Average of $x^2$ coefficient across datasets for quadratic fitting.

| Dataset | BZR | COX2 | MUTAG | REDDIT-5k |
|---|---|---|---|---|
| **Average of $x^2$ Coefficient** | $4.71 \times 10^{-5}$ | $6.61 \times 10^{-4}$ | $1.16 \times 10^{-2}$ | $1.78 \times 10^{-5}$ |

## B.4 Interpreting TopER

Our approach involves accurately modeling the evolution of a graph throughout the filtration process. One can easily identify clusters for each class and outliers in the other datasets given in Figure 1a and make inferences about the different clusters and outliers. Furthermore, when the pivot $a_f$ is positive or negative, it can be interpreted as graph density behavior in the filtration sequence (See Figure 7).

## C Proofs of Stability Theorems

In this part, we prove the stability results for our TopER.

**Lemma 1.** *Skraba and Turner (2020) Let $\mathcal{X}$ be a compact metric space, and $f, g : \mathcal{X} \to \mathbb{R}$ be two filtration functions. Then, for any $p \geq 1$, we have $\mathcal{W}_p(\mathrm{PD}_k(\mathcal{X}, f), \mathrm{PD}_k(\mathcal{X}, g)) \leq \|f - g\|_p$*

The next lemma is on the stability of Betti curves by Dłotko and Gurnari (2023) [Proposition 1].

**Lemma 2.** *Dłotko and Gurnari (2023) Let $\beta_k(\mathcal{X})$ is the $k^{th}$ Betti function obtained from the persistence module $\mathrm{PM}_k(\mathcal{X})$.*

$$\|\beta_k(\mathcal{X}) - \beta_k(\mathcal{Y})\|_1 \leq 2\mathcal{W}_1(\mathrm{PD}_k(\mathcal{X}), \mathrm{PD}_k(\mathcal{Y}))$$

Now, we are ready to prove our stability result.

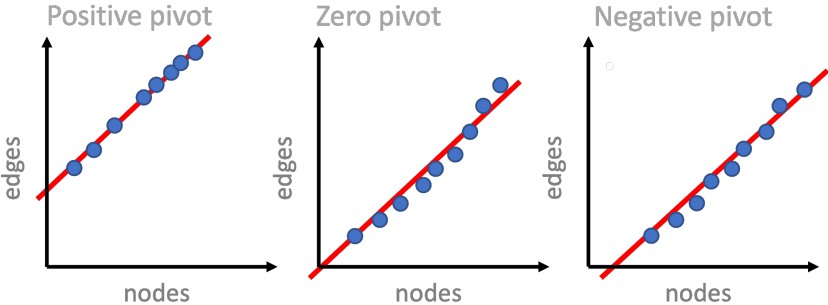

Figure 7: **Pivot Behavior.** A graph can exhibit three distinct pivot behaviors. Positive pivot graphs display a cluster of vertices that are closely interconnected and appear early in the filtration process. On the other hand, negative pivot graphs feature loosely connected nodes where the edges enter the filtration at a later stage. Graphs with zero pivot are usually quasi-complete graphs.

**Theorem 2.** *Let $\mathcal{X}$ be a compact metric space, and $f, g : \mathcal{X} \to \mathbb{R}$ be two filtration functions. Then, for some $C > 0$,*

$$\|\mathrm{TE}_f(\mathcal{X}) - \mathrm{TE}_g(\mathcal{X})\|_1 \leq C \cdot \mathcal{W}_1(\mathrm{PD}_k(\mathcal{X}, f), \mathrm{PD}_k(\mathcal{X}, g))$$

*Proof.* We will utilize the stability theorems from topological data analysis given above.

First, we employ the stability of Betti curves by Lemma 2.

$$\|\beta_k(\mathcal{X}) - \beta_k(\mathcal{Y})\|_1 \leq 2\mathcal{W}_1(\mathrm{PD}_k(\mathcal{X}), \mathrm{PD}_k(\mathcal{Y})) \tag{1}$$

Hence to obtain $\mathrm{TE}_f(\mathcal{X}) = (a_f, b_f)$, we fit least squares line $y = a_f + b_f x$ to the set of $N$ points in $\mathbb{R}^2$, i.e., $\mathcal{Z}_f = \{(\beta_0^f(\epsilon_i), \beta_1^f(\epsilon_i))\}_{i=1}^N$. Similarly, we obtain $\mathrm{TE}_g(\mathcal{X}) = (a_g, b_g)$ by fitting least squares line to $\mathcal{Z}_g = \{(\beta_0^g(\epsilon_i), \beta_1^g(\epsilon_i))\}_{i=1}^N$. By Equation (1), we have

$$\mathbf{D}_H(\mathcal{Z}_f, \mathcal{Z}_g) \leq 4\mathcal{W}_1(\mathrm{PD}_k(\mathcal{X}), \mathrm{PD}_k(\mathcal{Y})) \tag{2}$$

where $\mathbf{D}_H(\mathcal{Z}_f, \mathcal{Z}_g)$ represent Hausdorff distance between the point clouds $\mathcal{Z}_f$ and $\mathcal{Z}_g$ in $\mathbb{R}^2$.

Now, by the stability of least squares fit with respect to Hausdorff distance (Chernov et al. (2012) [Theorem 3.1]), we have

$$\|\mathrm{TE}_f(\mathcal{X}) - \mathrm{TE}_g(\mathcal{X})\|_1 \leq C \cdot \mathbf{D}_H(\mathcal{Z}_f, \mathcal{Z}_g) \tag{3}$$

Hence, when we combine Equations (2) and (3), we have

$$\|\mathrm{TE}_f(\mathcal{X}) - \mathrm{TE}_g(\mathcal{X})\|_1 \leq C \cdot \mathcal{W}_1(\mathrm{PD}_k(\mathcal{X}), \mathrm{PD}_k(\mathcal{Y}))$$

The proof follows. □

By combining the above result with Lemma 1, we obtain the following corollary.

**Corollary 2.** *Let $\mathcal{X}$ be a compact metric space, and $f, g : \mathcal{X} \to \mathbb{R}$ be two filtration functions. Then, for some $C > 0$,*

$$\|\mathrm{TE}_f(\mathcal{X}) - \mathrm{TE}_g(\mathcal{X})\|_1 \leq C \cdot \|f - g\|_1$$

*Proof.* By Lemma 1, we have

$$\mathcal{W}_1(\mathrm{PD}_k(\mathcal{X}, f), \mathrm{PD}_k(\mathcal{X}, g)) \leq \|f - g\|_1 \tag{4}$$

By Theorem 2, we have

$$\|\mathrm{TE}_f(\mathcal{X}) - \mathrm{TE}_g(\mathcal{X})\|_1 \leq C \cdot \mathcal{W}_1(\mathrm{PD}_k(\mathcal{X}, f), \mathrm{PD}_k(\mathcal{X}, g)) \tag{5}$$

By combining Equations (4) and (5), we conclude

$$\|\mathrm{TE}_f(\mathcal{X}) - \mathrm{TE}_g(\mathcal{X})\|_1 \leq \widehat{C} \cdot \|f - g\|_1$$

The proof follows. □

## D  Filtration Functions

In this section we will write the definitions of filtration functions we considered, except atomic weight and popularity (which is defined in the main text). Let us consider a graph $\mathcal{G} = (\mathcal{V}, \mathbb{E})$.

- The *degree* of a node $u$ refers to the number of edges incident to $u$ (Evans and Chen, 2022). More specifically, it is the size of the neighborhood of $u$, which is the set of nodes that are directly connected to $u$ by edges. The formal expression for the degree of node $u$ is given by:

$$\deg(u) = |\mathcal{N}(u)|$$

where:

- $\mathcal{N}(u)$ is the neighborhood of node $u$, which is the set of all nodes adjacent to $u$,
- $|N(u)|$ denotes the cardinality (size) of the set $N(u)$, which corresponds to the number of nodes directly connected to $u$ by edges.

- The *closeness* (Evans and Chen, 2022) function of a node $u$ is defined as:

$$\text{Closeness}(u) = \sum_{v \in \mathcal{V} \setminus \{u\}} \frac{1}{d(u, v)},$$

where:

- $\mathcal{V}$ is the set of all nodes in the graph,
- $d(u, v)$ is the shortest-path distance between nodes $u$ and $v$,
- $\mathcal{V} \setminus \{u\}$ denotes the set of all nodes in $\mathcal{V}$ except $u$.

- The *degree centrality* of a node $u$ is defined as the ratio of the degree of node $u$ to the total number of nodes in the network, excluding the node itself (Evans and Chen, 2022). The formal expression for *degree centrality* is given by:

$$\text{Degree Centrality}(u) = \frac{\deg(u)}{|V| - 1}$$

where:

- $\deg(u)$ is the degree of node $u$,
- $|\mathcal{V}| - 1$ represents the total possible connections a node can have, excluding itself.

-The *Forman-Ricci* curvature (Lin et al., 2011) for an edge $(u, v)$ is defined as:

$$F_{\text{Ricci}}(u, v) = 4 - (\deg(u) + \deg(v)) + |\mathcal{N}(u) \cap \mathcal{N}(v)|,$$

where:

- $\deg(u)$ and $\deg(v)$ are the degrees of nodes $u$ and $v$,
- $\mathcal{N}(u)$ and $\mathcal{N}(v)$ are the sets of neighbors of $u$ and $v$,
- $|\mathcal{N}(u) \cap \mathcal{N}(v)|$ is the size of the intersection of the neighbors of $u$ and $v$.

- The $\alpha$-weighted Ollivier Ricci (Lin et al., 2011) curvature between two nodes $x$ and $y$ is defined as:

$$\kappa_\alpha(x, y) = 1 - \frac{W_1(\mu_x^\alpha, \mu_y^\alpha)}{d(x, y)},$$

where:

- $W_1(\mu_x^\alpha, \mu_y^\alpha)$ is the Wasserstein-1 distance between the $\alpha$-weighted neighborhood distributions $\mu_x^\alpha$ and $\mu_y^\alpha$,
- $d(x, y)$ is the shortest-path distance between nodes $x$ and $y$.

- The $\alpha$-weighted neighborhood distribution $\mu_x^\alpha(z)$ for node $x$ is given by:

$$\mu_x^\alpha(z) = \begin{cases} \alpha & \text{if } z = y, \\ (1-\alpha)\frac{1}{\deg(x)} & \text{if } z \in \mathcal{N}(x), \\ 0 & \text{otherwise,} \end{cases}$$

where:

- $\deg(x)$ is the degree of node $x$,
- $\mathcal{N}(x)$ is the set of neighbors of $x$,
- $\alpha \in [0, 1]$ is a parameter balancing the focus on the direct edge to $y$ versus the uniform distribution over $x$'s neighbors.

Substituting $\mu_x^\alpha$ and $\mu_y^\alpha$ into the curvature formula, we have:

$$\kappa_\alpha(x, y) = 1 - \frac{\inf_{\pi \in \Pi(\mu_x^\alpha, \mu_y^\alpha)} \sum_{u \in N(x)} \sum_{v \in N(y)} \pi(u, v) d(u, v)}{d(x, y)},$$

where $\Pi(\mu_x^\alpha, \mu_y^\alpha)$ is the set of all joint probability distributions with marginals $\mu_x^\alpha$ and $\mu_y^\alpha$, and $d(u, v)$ is the distance between nodes $u$ and $v$.

