# OpenReview forum: "TopER: Topological Embeddings in Graph Representation Learning"
_ICLR.cc/2025/Conference — Submitted to ICLR 2025_

### Official Review · Reviewer_pfxt · 2024-10-28

**Soundness:** 3
**Presentation:** 4
**Contribution:** 4
**Rating:** 8
**Confidence:** 4

**Summary:**

This paper focuses on characterizing graph topological structures with persistent homology methods. The proposed TopER rerpesents graphs with a sereis of graph filtrations and characterizes graphs with merely two parameters. Experimental results validate the effectiveness of the proposed TopER in generating self-supervised graph representations.

**Strengths:**

+ A novel simple yet framework to represent graphs with persistence homology.
+ Compared with existing mainstream graph contrastive learning based method, the proposed TopER is parameter-free while achieving promising results.
+ Solid experiments and ablation studies on the proposed TopER.
+ The paper is well written and easy to follow.

**Weaknesses:**

- The reason of adapting linear regression to fit the filtration sequences is mentioned in the Appendix, yet no theoretical or experimental support are provided.
- The key parts of TopER's implementation are missing from the provided repo link.
- The used filtration functions require further introduction and explanation.

**Questions:**

1. The `functions_to_calculate_[a,b]` folder in the anonymous repository appears to be empty. Could you explain why?
2. Is there a theoretical basis for the linear correlation between $a$ and $b$ ? How would the model performance when applying other fitting functions, such as polynomials?

---

> ### Author Response · Authors · 2024-11-19
>
> *We thank the reviewer for their valuable feedback, which, along with comments from others, has guided significant improvements to our paper.*
>
> **W1. Linear Regression**
> >The reason of adapting linear regression to fit the filtration sequences is mentioned in the Appendix, yet no theoretical or experimental support are provided.
>
> Thank you for your insightful question. From a mathematical perspective, it is indeed possible to construct examples where higher-degree fitting functions outperform a linear fit. However, such cases are exceptional due to the following reasons:
>
> Let $x_i = |V_i|$ and $y_i = |E_i|$, where $G_i = (V_i, E_i)$ represents a subgraph in the filtration sequence {$G_i$} for $1\leq i\leq N$. The problem reduces to finding the best fit for $N$ points $Q = \{(x_i, y_i)\}_{i=1}^N$.
>
> ### Observations
> 1. **Monotonicity of Sequences**:
>    Both $\{x_i\}$ and $\{y_i\}$ are monotone non-decreasing sequences. By appropriately selecting thresholds and ignoring repetitions, these sequences can be assumed to be strictly monotone increasing. Consequently, $Q$ lies on the graph of an increasing function $y = f(x)$.
>
> 2. **Behavior of $f(x)$**:
>    The nature of $f(x)$ determines the fitting model:
>    - **Linear Fit (Order 1)**: $y = ax + b$
>    - **Quadratic Fit (Order 2)**: $y = ax^2 + bx + c$ (convex or concave)
>    - **Higher-Order Fits**: These correspond to cases where the curve exhibits mixed convex and concave behavior.
>
> The behavior of $f(x)$ depends on the rate of change, captured by the sequence $\Delta_i = \frac{y_{i+1} - y_i}{x_{i+1} - x_i}$:
>    - **Linear Fit**: $\Delta_i$ is approximately constant.
>    - **Quadratic Fit**: $\Delta_i$ is either monotonically increasing (convex) or decreasing (concave).
>    - **Higher-Order Fits**: $\Delta_i$ does not follow a monotone pattern.
>
> ### Experimental Insights
> In our experiments, we did not observe non-monotonic behavior in $\{\Delta_i\}$, eliminating the need for higher-order fits. Instead, we compared linear (order 1) and quadratic (order 2) fits. Interestingly, the linear fit consistently provided better approximations. This is because, for most quadratic fits, the coefficient of the quadratic term was negligibly small, indicating a near-linear relationship.
>
> To support these findings, we provide plots in **Appendix B.3**, illustrating the behavior of $f(x)$ under various fitting models. Furthermore, we conducted the following experiments to further validate the robustness of the linear fit.  In the table below, you can see the average of the coefficients of quadratic terms when we use quadratic fit for the datasets, i.e. if we fit $y=a+bx+cx^2$ polynomial, we observe that quadratic term is mostly negligible, and the tends to be a linear fit. We added this table to Appendix Section B.3.
>
> Here’s the transposed table:
>
> |              | BZR        | COX2       | MUTAG      | REDDIT-5k |
> |--------------|------------|------------|------------|-----------|
> | Dataset      | BZR        | COX2       | MUTAG      | REDDIT-5k |
> | Average of $x^2$ Coefficient | $4.71 \times 10^{-5}$ | $6.61 \times 10^{-4}$ | $1.16 \times 10^{-2}$ | $1.78 \times 10^{-5}$ |
>
> **W2. Missing parts of Repo**
> >The key parts of TopER's implementation are missing from the provided repo link.
>
> We apologize for the problem. Due to the blind submission requirement, we used an anonymous GitHub link to share our code. However, for some reason, while the repository exists in the folders, it does not display correctly via the anonymous link at https://anonymous.4open.science/r/TopER-AA38
>
> However, the original repo can be viewed at https://github.com/AstroComets/TopER
>
> **W3. Filtration function descriptions**
> >The used filtration functions require further introduction and explanation.
>
> Thank you for this feedback. We added detailed descriptions of filtration functions to Appendix D.
>
> **Q1. Repository empty**
> >The functions_to_calculate_[a,b] folder in the anonymous repository appears to be empty. Could you explain why?
>
> Answered above.
>
> **Q2. Correlation between a and b?**
> > Is there a theoretical basis for the linear correlation between a and b? How would the model performance when applying other fitting functions, such as polynomials?
>
> Answered above.
>
> *Thank you once again for your valuable feedback. We are happy to address any further questions or comments you may have.*

---

### Official Review · Reviewer_YMi9 · 2024-11-02

**Soundness:** 2
**Presentation:** 2
**Contribution:** 2
**Rating:** 5
**Confidence:** 4

**Summary:**

This paper presents Topological Evolution Rate (TopER), a novel low-dimensional embedding method based on topological data analysis for intuitive and interpretable graph representations. TopER achieves competitive performance in graph clustering and classification, with strong results across diverse datasets.

**Strengths:**

1. TopER introduces a topologically grounded, low-dimensional embedding approach that efficiently captures graph substructure evolution, simplifying the computation-heavy process of Persistent Homology.

2. TopER provides clear, low-dimensional embeddings that highlight key topological features such as clusters and outliers, enhancing the ability to analyze individual and cross-dataset graphs.

**Weaknesses:**

1. The methodology briefly mentions alternatives to linear fitting, such as polynomial fits, but lacks detailed explanations. This omission raises questions about the impact of different fitting methods on the TopER vector and its adaptability to diverse graph types.
2. Although the linear regression on filtration sequences is computationally simpler than full Persistent Homology, the paper does not clarify how this process scales for large graphs with numerous filtration levels, making it uncertain whether the approach maintains efficiency for complex, high-dimensional data.
3. The ablation study shows that combining different filtration functions improves performance. However, it does not clearly explain how each function helps or why they work well together. This makes it hard to know which functions are truly needed and which may just add extra complexity without much benefit.

**Questions:**

Can the authors clarify why combining these specific filtration functions enhances performance in ablation study?

---

> ### Author Response · Authors · 2024-11-19
>
> *We thank the reviewer for their valuable feedback, which, along with comments from others, has guided significant improvements to our paper. We hope our revisions and responses merit reconsideration of a higher evaluation.*
>
> **W1. Lack of Threshold Setting Details:**
> >The paper does not explain how thresholds are set for different filtration functions, nor how the number of subgraphs n is determined. Since both threshold choice and n could impact results, further analysis would be valuable.
>
> Thank you for this question. We had used 500 thresholds to capture finer-grained information, as the model is computationally efficient and adding more thresholds incurs minimal additional cost. Based on your review, we experimented with more thresholds and found that TopER remains robust and highly effective. We have added this table to Appendix Section A.4.
>
> | **Number of Thresholds** | **PROTEINS**      | **REDDIT-B**     | **REDDIT-5K**     |
> |---------------------------|-------------------|------------------|-------------------|
> | 10                        | 72.78 ± 4.04     | 90.55 ± 1.96     | 55.99 ± 1.97      |
> | 20                        | 74.31 ± 3.23     | 91.20 ± 1.66     | 55.91 ± 2.14      |
> | 50                        | 74.76 ± 4.55     | 92.05 ± 1.96     | 55.39 ± 2.10      |
> | 100                       | 73.85 ± 3.67     | 92.85 ± 1.18     | 55.51 ± 2.61      |
> | 200                       | 75.47 ± 3.06     | 93.15 ± 2.10     | 56.51 ± 2.04      |
> | 500                       | 74.58 ± 3.92      | 92.70 ± 2.38     | 56.51 ± 2.22      |
>
> **W2. Clarity on Embedding Dimension:**
> >The paper lacks details on TopER’s final embedding dimension and does not directly compare it to the embedding dimensions of other algorithms. Embedding dimension is crucial for evaluating performance and computational efficiency, so specifying and comparing the dimensions of different methods would enhance the clarity and interpretability of the results.
>
> Thank you for raising this important question. In our experiments, we used the same six filtration functions across all datasets, employing both sublevel and superlevel filtrations, along with an additional atomic weight filtration function specifically for molecular datasets. Each filtration function produces a 2-dimensional TopER embedding, resulting in a total of 24 dimensions per dataset (6 functions× 2 filtrations x 2) or 28 dimensions for molecular datasets. To optimize performance and reduce redundancy, we applied a feature selection algorithm as part of our ML pipeline to identify the most informative features while mitigating the impact of correlated dimensions. The final dimensions after feature selection are given below. We have also incorporated this table into Table 7 (Hyperparameters table).
>
> For the embedding dimensions of other methods, vanilla persistent homology embeddings typically correspond to the number of thresholds used in the filtration step. For example, the degree function often results in 10–20 dimensions, while more complex functions like betweenness or closeness centrality can yield 200–300 dimensions. In contrast, recent GNNs generally use embedding dimensions of 48,  64, 128, or even 512, depending on the pooling mechanism. Therefore, TopER operates with significantly lower embedding dimensions compared to these baseline methods, highlighting its computational efficiency.
>
> Details of the TopER embeddings and the feature selection methodology are provided in Section 5.1. Furthermore, the individual performance of each filtration function (based solely on 2-dimensional embeddings) is reported in Table 6, offering insights into the contributions of specific filtration strategies.
>
> **Dataset**         | BZR  | COX2 | MUTAG | PROTEINS | IMDB-B | IMDB-M | REDDIT-B | REDDIT-5K |
> |---------------------|------|------|-------|----------|--------|--------|----------|-----------|
> | **TopER Dimension** | 26   | 26   | 20    | 26       | 20     | 20     | 24       | 14        |

---

> ### Author Response · Authors · 2024-11-19
>
> **W3. Dataset-Specific Dimension Variability:**
> >Line 394 mentions "top-performing combinations of filtration and vectorization for each dataset," which suggests that embedding dimensions might vary across datasets. However, the paper does not clarify what these combinations are or discuss the impact of using different combinations across datasets, especially given the many possible configurations with the eight filtration functions.
>
> Thank you for raising this valid concern. As detailed in the previous response, we start with the same seven filtration functions for all datasets, as detailed in Section 5.1. Our framework chooses the most informative filtrations which we report below. As our results in Table 6 of the article results, TopER performance with individual filtrations is comparable with SOTA and all TopER with 7 filtrations exceed SOTA results.
>
> Additionally, to assess the impact of embedding dimensions, we conducted new experiments evaluating the performance of the TopER model by progressively adding each filtration function step by step. This analysis, reported below,  provides insights into how the inclusion of additional filtration functions influences the model's performance. We include this table along with corresponding filtration functions in Appendix Section A.6.
>
> | *Dataset*   | *TopER-1*         | *TopER-2*         | *TopER-3*         | *TopER-4*         |
> |---------------|---------------------|---------------------|---------------------|---------------------|
> | *BZR*       | 82.48 ± 1.98        | 84.70 ± 2.84        | 85.66 ± 5.00        | 86.68 ± 3.81        |
> | *COX2*      | 78.81 ± 1.94        | 79.26 ± 4.86        | 79.04 ± 7.49        | 80.30 ± 3.91        |
> | *MUTAG*     | 86.14 ± 6.38        | 88.33 ± 3.88        | 86.75 ± 4.78        | 88.30 ± 4.63        |
> | *PROTEINS*  | 74.03 ± 2.71        | 74.67 ± 2.73        | 75.21 ± 3.39        | 75.65 ± 3.87        |
> | *IMDB-B*    | 73.00 ± 4.40        | 74.20 ± 4.26        | 74.50 ± 3.50        | 74.70 ± 3.95        |
> | *IMDB-M*    | 48.73 ± 4.33        | 49.80 ± 2.94        | 49.73 ± 4.18        | 49.87 ± 4.00        |
> | *REDDIT-B*  | 81.95 ± 2.74        | 90.45 ± 2.55        | 91.05 ± 2.62        | 91.50 ± 2.01        |
> | *REDDIT-5K* | 50.21 ± 1.41        | 54.11 ± 2.43        | 56.19 ± 2.40        | 56.33 ± 2.74        |
>
> **W4. Contribution of Each filtration:**
> >The ablation study does not analyze the effects of each filtration function on individual datasets, missing insights into how specific functions influence results across different types of data.
>
> Thank you for your insightful feedback. We would like to draw your attention to Table 6, which presents the impact of each filtration function on individual datasets. This table aims to provide a detailed analysis of how specific filtration functions influence the results across various types of data. If there are specific aspects you believe we have overlooked, we would be happy to provide further clarification or expand on this analysis in the revision.
>
> *Thank you once again for your valuable feedback. We are happy to address any further questions or comments you may have.*

---

> ### Comment · Reviewer_YMi9 · 2024-11-23
>
> Thank you for all the provided replies, and some of my concerns are addressed. I will keep my rating as 5.

---

> > ### Author Response · Authors · 2024-11-23
> >
> > Thank you once again for your insightful feedback and valuable time. If there are any further points you would like us to clarify, we would be more than happy to address them.

---

> > ### Author Response · Authors · 2024-11-29
> >
> > Dear Reviewer YMi9,
> >
> > Thank you once again for your insightful feedback and the time you’ve dedicated to reviewing our work. With the discussion period now extended, we would greatly appreciate it if you could share any remaining questions or concerns that you feel have not been addressed. We would be more than happy to provide further clarification.

---

### Official Review · Reviewer_QF1b · 2024-11-02

**Soundness:** 3
**Presentation:** 3
**Contribution:** 2
**Rating:** 6
**Confidence:** 3

**Summary:**

The paper presents a novel approach to train low-dimensional graph embeddings inspired by topological graph data analysis. Their method, Topological Evolution Rate (TopER), argues that the graph structure can be characterized by the linear function that relates how the number of edges relates to the number of nodes in the graphs induced by the nodes/edges that have a given value of a property such as degree. Through experiments, they show superior performance in graph classification, clustering, and interpretability. Furthermore, their method enables embedding different graph datasets in a shared embedding space, which is interesting.

**Strengths:**

- The paper is well-written with minimal typos and gives a good background of topological learning.
- Experiments are mostly thorough and provide a comparison against different baselines, especially for graph classification.
- The ability to train a shared interpretable embedding space of different graph datasets is interesting and has a lot of potential impact.
- The proposed method is computationally efficient and shows significant gains in performance.

**Weaknesses:**

- Motivation behind the method is lacking: The paper provides very little motivation behind the actual methodology.
  - It is not clear what the correlation between the number of edges and number of nodes of the induced graphs wrt a filtration function has to do with how the graph is characterized/classified.
  - It is not clearly stated why there should be a linear correlation between the two numbers. The motivating figures also do not motivate a linear correlation, except for MUTAG.
  - If there is a relation with persistence homology, then it should be discussed.
- The method is limited to non-attributed graphs and cannot easily extend to attributed graphs, for which it might depend on other graph kernels to be devised.
- High sensitivity to filtration functions: As can be noted by the ablation study, the performance is highly sensitive to the filtration functions used for a particular dataset. This limits the strengths since it is not clear if the set of filtration functions considered in this work are even exhaustive.
- Training an MLP on top of (a, b) of different filtration functions also limits the interpretability since the discriminative features may be formed as a combination of different filtration trends. Furthermore, using a single filtration function is always less performative than using multiple functions, which means the evolution rate of no one function is capable of classifying the graphs accurately.
- A simpler comparison should be provided that compares persistent homology and TopER in their time complexity and graph classification accuracy assuming the same filtration function.
- Hyperparameter analysis of the number of thresholds is missing.
- Empirical comparison with some important baselines is not provided:
  - Immonen, Johanna, Amauri Souza, and Vikas Garg. "Going beyond persistent homology using persistent homology." Advances in Neural Information Processing Systems 36 (2024).
  - Hofer, Christoph, et al. "Graph filtration learning." International Conference on Machine Learning. PMLR, 2020.
  - Rieck, Bastian, Christian Bock, and Karsten Borgwardt. "A persistent weisfeiler-lehman procedure for graph classification." International Conference on Machine Learning. PMLR, 2019.

**Questions:**

See above weaknesses.

---

> ### Author Response · Authors · 2024-11-19
>
> *We thank the reviewer for their valuable feedback, which, along with comments from others, has guided significant improvements to our paper. We hope our revisions and responses merit reconsideration of a higher evaluation.*
>
> **W1. Motivation behind the method is lacking:**
> >The paper provides very little motivation behind the actual methodology.
>
> Thank you for raising this concern. Our motivation is twofold:
>
> 1. Simplifying persistent homology methods in graph representation learning.
> 2. Addressing the significant gap in graph dataset visualization and low-dimensional graph embeddings.
>
> *First goal - why is it important?:* Persistent homology extracts features by recording the topological changes across filtration sequences of the data, and this approach has shown significant success in various settings. However, the approach is not feasible due to large computational costs.
>
> *First goal - how does TopER solve it?:* We simplify and streamline the filtration process by using an evolutionary rate extracted through domain-agnostic, computationally cheap functions.
>
> *Second goal - why is it important?:* While there has been considerable progress in node classification and visualization through node embeddings, there are virtually no methods, apart from Spectral Zoo, for graph classification visualization via graph embeddings.
>
> *Second goal - how does TopER solve it?:* TopER embeddings summarize graph filtration and create visually distinct and interpretable 2D representations by leveraging simple regression on the filtration process. This approach allows for direct visualization of graph structures, enabling the identification of clusters, outliers, and comparative patterns across different datasets.
>
> We have described these aspects in the introduction, but we agree that Section 4 could be an ideal place to elaborate on our contribution. We will change the section accordingly during our final revision.
>
> **W1.1 Method reasoning**
> > It is not clear what the correlation between the number of edges and number of nodes of the induced graphs wrt a filtration function has to do with how the graph is characterized/classified.
>
> Thank you for raising this point. We believe that topology is an excellent match for graph level tasks (as we outlined in page 2), as the field focuses on studying the properties of spaces and data that remain invariant under continuous deformations. Our motivation stems from the significant success of persistent homology (PH) in the past decade. In essence, PH tracks topological changes in a sequence of simplicial complexes induced by subgraphs $G_1 \subset G_2 \subset \dots \subset G_N$, which are generated based on a filtration function.
>
> For example, consider the degree function as a filtration function. In this case, the lower-index subgraphs consist of nodes with low degrees—representing "less important" nodes in terms of degree—while higher-index subgraphs progressively activate nodes of greater importance according to the degree hierarchy. This filtration process provides a structured decomposition of the graph, where the filtration function organizes nodes and edges into a hierarchy of importance.
>
> The sequence of subgraphs $\{G_i\}$ thus reveals how nodes and edges are distributed and connected with respect to this hierarchical structure. **TopER (Topological Evolution Rate)** captures this evolution pattern, quantifying how the graph's structure evolves as dictated by the filtration function. When the filtration function is relevant to the graph's domain—such as atomic weight in molecular graphs or transaction amounts in financial graphs—this evolution rate becomes both effective and interpretable.
>
> Our visualizations demonstrate the utility of this approach, showing how this simple idea can effectively detect clusters and outliers in graph datasets. Additionally, by combining TopER features derived from multiple filtration functions, our model achieves performance comparable to state-of-the-art methods on benchmark graph classification datasets.
>
> We believe this perspective clarifies the relationship between the evolution of node and edge distributions in the filtration process and the characterization of graphs in our method.

---

> ### Author Response · Authors · 2024-11-19
>
> **W1.2 Why linear**
> >It is not clearly stated why there should be a linear correlation between the two numbers. The motivating figures also do not motivate a linear correlation, except for MUTAG.
>
> Thank you for your insightful question. From a mathematical perspective, it is indeed possible to construct examples where higher-degree fitting functions outperform a linear fit. However, such cases are exceptional due to the following reasons:
>
> Let $x_i = |V_i|$ and $y_i = |E_i|$, where $G_i = (V_i, E_i)$ represents a subgraph in the filtration sequence {$G_i$} for $1\leq i\leq N$. The problem reduces to finding the best fit for $N$ points $Q = \{(x_i, y_i)\}_{i=1}^N$.
>
> ### Observations
> 1. **Monotonicity of Sequences**:
>    Both $\{x_i\}$ and $\{y_i\}$ are monotone non-decreasing sequences. By appropriately selecting thresholds and ignoring repetitions, these sequences can be assumed to be strictly monotone increasing. Consequently, $Q$ lies on the graph of an increasing function $y = f(x)$.
>
> 2. **Behavior of $f(x)$**:
>    The nature of $f(x)$ determines the fitting model:
>    - **Linear Fit (Order 1)**: $y = ax + b$
>    - **Quadratic Fit (Order 2)**: $y = ax^2 + bx + c$ (convex or concave)
>    - **Higher-Order Fits**: These correspond to cases where the curve exhibits mixed convex and concave behavior.
>
> The behavior of $f(x)$ depends on the rate of change, captured by the sequence $\Delta_i = \frac{y_{i+1} - y_i}{x_{i+1} - x_i}$:
>    - **Linear Fit**: $\Delta_i$ is approximately constant.
>    - **Quadratic Fit**: $\Delta_i$ is either monotonically increasing (convex) or decreasing (concave).
>    - **Higher-Order Fits**: $\Delta_i$ does not follow a monotone pattern.
>
> ### Experimental Insights
> In our experiments, we did not observe non-monotonic behavior in $\{\Delta_i\}$, eliminating the need for higher-order fits. Instead, we compared linear (order 1) and quadratic (order 2) fits. Interestingly, the linear fit consistently provided better approximations. This is because, for most quadratic fits, the coefficient of the quadratic term was negligibly small, indicating a near-linear relationship.
>
> To support these findings, we provide plots in **Appendix B.3**, illustrating the behavior of $f(x)$ under various fitting models. Furthermore, we conducted the following experiments to further validate the robustness of the linear fit.  In the table below, you can see the average of the coefficients of quadratic terms when we use quadratic fit for the datasets, i.e. if we fit $y=a+bx+cx^2$ polynomial, we observe that quadratic term is mostly negligible, and the tends to be a linear fit.
>
> | Dataset      | Average of x^2 Coefficient  |
> |--------------|------------------------------|
> | BZR          | 4.71 × 10⁻⁵                 |
> | COX2         | 6.61 × 10⁻⁴                 |
> | MUTAG        | 1.16 × 10⁻²                 |
> | REDDIT-5k    | 1.78 × 10⁻⁵                 |
>
> **W1.3 Relation with PH**
> >If there is a relation with persistence homology, then it should be discussed.
>
> Thank you for this question. In Section 3, we discussed the PH framework and emphasized the central role of filtration in this context. Following this foundation, in Section 4, we adopted the same approach as PH up to the filtration step, where a sequence of subgraphs $G_1 \subset G_2 \subset \dots \subset G_N$ is generated. However, our methodology diverges in the vectorization step, as we take a more direct approach to summarize the evolution pattern within the filtration sequence.  In revision, we added a remark in Section 4 (before subsection 4.1) to clarify this connection.

---

> ### Author Response · Authors · 2024-11-19
>
> **W2. Attributed graphs?**
> >The method is limited to non-attributed graphs and cannot easily extend to attributed graphs, for which it might depend on other graph kernels to be devised.
>
> Thank you very much for this insightful question. For attributed graphs, node or edge attributes can be effectively leveraged to define filtration functions. In our experiments, we utilized this approach for molecular graphs, where node attributes such as atomic weights provided a meaningful filtration function. Similarly, for transaction networks, one might use transaction amounts as edge filtration functions, and for distribution networks, voltage values could serve as node filtration functions.
>
> When the node attributes are high-dimensional, dimension reduction techniques like tSNE or alphacore (a data depth-based method) can be employed to derive filtration functions that capture the most relevant information. Among our datasets, only BZR and COX2 had high-dimensional node attributes. For these datasets, we used tSNE and alphacore to induce filtration functions, ensuring that the extracted features were both meaningful and computationally efficient. We report the results of these experiments below.
>
>
> | Dataset | TSNE1         | TSNE2         | TSNE1+2           | alphacore      | TopER+TSNE     | TopER+alphacore |
> |---------|---------------|---------------|----------------|----------------|----------------|-----------------|
> | BZR     | 80.02 ± 4.06  | 79.26 ± 1.93  | 81.27 ± 4.98   | 81.49 ± 3.29   | 87.92 ± 3.32   | 88.89 ± 2.29    |
> | COX2    | 76.21 ± 4.84  | 73.44 ± 4.97  | 77.30 ± 3.96   | 68.96 ± 8.39   | 81.79 ± 3.76   | 81.37 ± 4.97    |
>
>
> **W3. High sensitivity to filtration functions:**
> >As can be noted by the ablation study, the performance is highly sensitive to the filtration functions used for a particular dataset. This limits the strengths since it is not clear if the set of filtration functions considered in this work are even exhaustive.
>
> Thank you for your insightful comment. We appreciate the reviewer's comment on the sensitivity of performance to different filtration functions. Our method is designed to use multiple filtrations to gain different perspectives on the data's structure. Consistent results across individual functions were not our focus. Instead, we aimed to combine their strengths for better outcomes.
>
> For example, degree and closeness filtrations show a high similarity, with a correlation of 0.96. On the other hand, the correlation between degree and Ricci curvature is 0.7, showing that these filtrations bring unique insights. This diversity is a strength of our method, allowing it to capture varied aspects of the data.
>
> As shown in Table 6, TopER's performance when combining these functions is better than using any one function alone. This demonstrates that using multiple filtrations together enhances the method's overall effectiveness- TopER reaches SOTA results.We also note that the individual performance of filtration functions is crucial for visualization, as better performance leads to clearer separation among classes.

---

> ### Author Response · Authors · 2024-11-19
>
> **W4. Interpretability**
> >Training an MLP on top of (a, b) of different filtration functions also limits the interpretability since the discriminative features may be formed as a combination of different filtration trends. Furthermore, using a single filtration function is always less performative than using multiple functions, which means the evolution rate of no one function is capable of classifying the graphs accurately.
>
> Thank you for your insightful comment. We acknowledge that combining multiple filtration functions can reduce interpretability, as the discriminative features may arise from a complex combination of trends from different filtrations. However, we believe this concern does not detract from the interpretability or performance of our approach in the contexts presented in our paper.
>
> In our work, we apply the TopER model in two distinct ways. The first approach utilizes a single filtration function to generate low-dimensional embeddings, which we use for visualizations. This approach provides highly interpretable results, as the embedding captures the evolution of the graph based on a single, well-defined filtration. Specifically, the parameters from the linear fit $y = ax + b$ (where $a$ represents growth and $b$ is the pivot) are directly interpretable. These parameters summarize the distribution of nodes and edges and the connectivity of the subgraphs induced by the hierarchy of the chosen filtration function. For instance, a large $a$ indicates high connectivity in relation to the filtration, and a negative $b$ suggests low connectivity in the induced subgraphs. We provide detailed illustrations and explanations of these relationships in the main text and in Appendix B.4.
>
> The second approach, involving multiple filtration functions, is used for more complex tasks like graph classification. While we agree that combining different filtrations may reduce the direct interpretability of individual features, it leads to SOTA performance. In our experiments, we show that this synergy between filtration functions improves the model’s ability to capture diverse structural patterns, thus enhancing classification accuracy.
>
> Moreover, many real-world datasets come with their own natural filtration functions—such as transaction amounts for financial networks, density for traffic networks, and power flow for distribution networks—that define meaningful hierarchies among nodes and edges. These domain-specific filtrations further enhance interpretability, as they provide context that aids in understanding the relationships within the data. By leveraging these naturally occurring filtration functions, our model not only achieves strong performance but also remains adaptable to real-world datasets, allowing for effective visualization, cluster detection, and outlier identification.
>
> We believe this dual approach offers a balance between interpretability and performance, and we have demonstrated its effectiveness across multiple domains.
>
> **W5. Time complexity PH vs. TopER**
> >A simpler comparison should be provided that compares persistent homology and TopER in their time complexity and graph classification accuracy assuming the same filtration function.
>
> Thank you very much for this feedback. In response to your suggestion, we conducted experiments comparing persistent homology and TopER using the same filtration function, specifically the sublevel degree filtration. For PH, we employed Betti vectorization. Our results, as shown below, demonstrate that TopER is significantly faster than PH.
>
> | **Dataset**  | **TopER-1 Time**  | **TopER-1 Accuracy**  | **PH Time**         | **PH Accuracy**     | **Number of Thresholds** |
> |--------------|-------------------|-----------------------|---------------------|---------------------|--------------------------|
> | BZR          | 3.13 s           | 82.73 ± 2.12          | 5.99 s             | 83.70 ± 3.51        | 4                        |
> | IMDB-B       | 16.52 s          | 73.10 ± 4.18          | 319.95 s      | 71.00 ± 4.07        | 65                       |
> | REDDIT-B     | 11 min 40.60 s   | 79.55 ± 2.20          | 152 min 53.37 s | 84.50 ± 2.51        | 501

---

> ### Author Response · Authors · 2024-11-19
>
> **W6. Number of Thresholds**
> >Hyperparameter analysis of the number of thresholds is missing.
>
> Thank you for this excellent question. In our experiments, we opted to use a large number of thresholds to capture finer-grained information, as the model is computationally efficient and adding more thresholds incurs minimal additional cost. That said, we also evaluated the model's performance with fewer thresholds and found that it remains robust and highly effective. We have added this table to Appendix Section A.4.
>
> Hyperparameter analysis of the number of thresholds on the number of thresholds.
>
> | **# Thresholds** | **PROTEINS**      | **REDDIT-B**     | **REDDIT-5K**     |
> |---------------------------|-------------------|------------------|-------------------|
> | 10                        | 72.78 ± 4.04     | 90.55 ± 1.96     | 55.99 ± 1.97      |
> | 20                        | 74.31 ± 3.23     | 91.20 ± 1.66     | 55.91 ± 2.14      |
> | 50                        | 74.76 ± 4.55     | 92.05 ± 1.96     | 55.39 ± 2.10      |
> | 100                       | 73.85 ± 3.67     | 92.85 ± 1.18     | 55.51 ± 2.61      |
> | 200                       | 75.47 ± 3.06     | 93.15 ± 2.10     | 56.51 ± 2.04      |
> | 500                       | 74.58 ± 3.92      | 92.70 ± 2.38     | 56.51 ± 2.22      |
>
> **W7. Missing baselines:**
> >-Immonen, Johanna, Amauri Souza, and Vikas Garg. "Going beyond persistent homology using persistent homology." Advances in Neural Information Processing Systems 36 (2024).
> -Hofer, Christoph, et al. "Graph filtration learning." International Conference on Machine Learning. PMLR, 2020.
> -Rieck, Bastian, Christian Bock, and Karsten Borgwardt. "A persistent weisfeiler-lehman procedure for graph classification." International Conference on Machine Learning. PMLR, 2019.
>
> Thank you for bringing these important references to our attention. Since all of these studies used the same experimental setup, we have included their reported results in our accuracy table (Table 2).
>
> *Thank you once again for your valuable feedback. We are happy to address any further questions or comments you may have.*

---

> > ### Comment · Reviewer_QF1b · 2024-11-22
> >
> > I thank the authors for providing a very comprehensive and detailed rebuttal. They have addressed most of my concerns and the contributions are more clear and empirically supported now. Thus, I have raised my score accordingly. However, I still feel the trade-off between interpretability and performance with respect to the number of filtration functions exists and should be addressed to some extent in this paper. Secondly, I believe Table 2 should also be completely filled for all the baselines by reproducing them on their datasets for a fair comparison.

---

> > > ### Author Response · Authors · 2024-11-23
> > >
> > > Thank you for your feedback and for raising your score. We will address the trade-off between interpretability and performance in the revised paper and complete Table 2 for all baselines to ensure a fair comparison. Thank you for your constructive suggestions, which will help improve the quality of our work.

---

### Official Review · Reviewer_4YJQ · 2024-11-03

**Soundness:** 2
**Presentation:** 3
**Contribution:** 2
**Rating:** 6
**Confidence:** 3

**Summary:**

This paper presents TopER, an innovative method for graph embedding based on topological data analysis (TDA). TopER introduces an efficient approach to graph representation learning by simplifying Persistent Homology, particularly in creating embeddings that capture the evolution of graph substructures. Instead of traditional graph neural network (GNN)-based methods, TopER focuses on generating interpretable, low-dimensional embeddings through filtration and linear regression on subgraph evolution rates. It shows competitive performance in clustering and classification tasks across various datasets, such as molecular and social networks, underscoring its utility for interpretable, scalable graph embeddings.

**Strengths:**

Innovative Approach: The paper introduces TopER, a novel method based on Topological Data Analysis (TDA) that simplifies Persistent Homology for graph embedding. By tracking the evolution rate of graph substructures, TopER provides a low-dimensional, interpretable graph representation, enabling it to capture graph structure effectively without high computational costs.
ss
Computational Efficiency: The limitations in computational efficiency are crucial since they directly impact the scalability of TopER for large real-world applications, a primary goal of embedding methods. Improving this area could make TopER more viable for large-scale datasets in practice.

Good Interpretability and Visualization: The two-dimensional embeddings produced by TopER offer strong interpretability, allowing users to easily identify clusters, outliers, and structural features within datasets. Unlike higher-dimensional embeddings, which are often challenging to interpret, TopER is well-suited for data exploration and graph visualization.

**Weaknesses:**

Lack of Threshold Setting Details: The paper does not explain how thresholds are set for different filtration functions, nor how the number of subgraphs n is determined. Since both threshold choice and n could impact results, further analysis would be valuable.

Clarity on Embedding Dimension: The paper lacks details on TopER’s final embedding dimension and does not directly compare it to the embedding dimensions of other algorithms. Embedding dimension is crucial for evaluating performance and computational efficiency, so specifying and comparing the dimensions of different methods would enhance the clarity and interpretability of the results.

Dataset-Specific Dimension Variability: Line 394 mentions "top-performing combinations of filtration and vectorization for each dataset," which suggests that embedding dimensions might vary across datasets. However, the paper does not clarify what these combinations are or discuss the impact of using different combinations across datasets, especially given the many possible configurations with the eight filtration functions.

Ablation Study Needs More Analysis: The ablation study does not analyze the effects of each filtration function on individual datasets, missing insights into how specific functions influence results across different types of data.

**Questions:**

See from Paper Weaknesses.

---

> ### Author Response · Authors · 2024-11-19
>
> *We thank the reviewer for their valuable feedback, which, along with comments from others, has guided significant improvements to our paper. We hope our revisions and responses merit reconsideration of a higher evaluation.*
>
> **W1. Lack of Threshold Setting Details:**
> >The paper does not explain how thresholds are set for different filtration functions, nor how the number of subgraphs n is determined. Since both threshold choice and n could impact results, further analysis would be valuable.
>
> Thank you for this question. We had used 500 thresholds to capture finer-grained information, as the model is computationally efficient and adding more thresholds incurs minimal additional cost. Based on your review, we experimented with more thresholds and found that TopER remains robust and highly effective. We have added this table to Appendix Section A.4.
>
>
> | **# Thresholds** | **PROTEINS**      | **REDDIT-B**     | **REDDIT-5K**     |
> |---------------------------|-------------------|------------------|-------------------|
> | 10                        | 72.78 ± 4.04     | 90.55 ± 1.96     | 55.99 ± 1.97      |
> | 20                        | 74.31 ± 3.23     | 91.20 ± 1.66     | 55.91 ± 2.14      |
> | 50                        | 74.76 ± 4.55     | 92.05 ± 1.96     | 55.39 ± 2.10      |
> | 100                       | 73.85 ± 3.67     | 92.85 ± 1.18     | 55.51 ± 2.61      |
> | 200                       | 75.47 ± 3.06     | 93.15 ± 2.10     | 56.51 ± 2.04      |
> | 500                       | 74.58 ± 3.92      | 92.70 ± 2.38     | 56.51 ± 2.22      |
>
>
> **W2. Clarity on Embedding Dimension:**
> >The paper lacks details on TopER’s final embedding dimension and does not directly compare it to the embedding dimensions of other algorithms. Embedding dimension is crucial for evaluating performance and computational efficiency, so specifying and comparing the dimensions of different methods would enhance the clarity and interpretability of the results.
>
> Thank you for raising this important question. In our experiments, we used the same six filtration functions across all datasets, employing both sublevel and superlevel filtrations, along with an additional atomic weight filtration function specifically for molecular datasets. Each filtration function produces a 2-dimensional TopER embedding, resulting in a total of 24 dimensions per dataset (6 functions× 2 filtrations x 2) or 28 dimensions for molecular datasets. To optimize performance and reduce redundancy, we applied a feature selection algorithm as part of our ML pipeline to identify the most informative features while mitigating the impact of correlated dimensions. The final dimensions after feature selection are given below. We have also incorporated this table into Table 7 (Hyperparameters table).
>
> For the embedding dimensions of other methods, vanilla persistent homology embeddings typically correspond to the number of thresholds used in the filtration step. For example, the degree function often results in 10–20 dimensions, while more complex functions like betweenness or closeness centrality can yield 200–300 dimensions. In contrast, recent GNNs generally use embedding dimensions of 48,  64, 128, or even 512, depending on the pooling mechanism. Therefore, TopER operates with significantly lower embedding dimensions compared to these baseline methods, highlighting its computational efficiency.
>
> Details of the TopER embeddings and the feature selection methodology are provided in Section 5.1. Furthermore, the individual performance of each filtration function (based solely on 2-dimensional embeddings) is reported in Table 6, offering insights into the contributions of specific filtration strategies.
>
> We appreciate your interest in the embedding dimensions and their role in the analysis. If further clarification or additional comparisons to the embedding dimensions of other methods would be helpful, we would be glad to provide them.
>
> **Dataset**         | BZR  | COX2 | MUTAG | PROTEINS | IMDB-B | IMDB-M | REDDIT-B | REDDIT-5K |
> |---------------------|------|------|-------|----------|--------|--------|----------|-----------|
> | **TopER Dimension** | 26   | 26   | 20    | 26       | 20     | 20     | 24       | 14        |

---

> ### Author Response · Authors · 2024-11-19
>
> **W3. Dataset-Specific Dimension Variability:**
> > Line 394 mentions "top-performing combinations of filtration and vectorization for each dataset," which suggests that embedding dimensions might vary across datasets. However, the paper does not clarify what these combinations are or discuss the impact of using different combinations across datasets, especially given the many possible configurations with the eight filtration functions.
>
> Thank you for raising this valid concern. As detailed in the previous response, we start with the same seven filtration functions for all datasets, as detailed in Section 5.1. Our framework chooses the most informative filtrations. As our results in Table 6 of the article results, TopER performance with individual filtrations is comparable with SOTA and all TopER with 7 filtrations exceed SOTA results.
> Additionally, to assess the impact of embedding dimensions, we conducted new experiments evaluating the performance of the TopER model by progressively adding each filtration function step by step. This analysis, reported below,  provides insights into how the inclusion of additional filtration functions influences the model's performance. We include the following table along with the filtration functions in Appendix Section A.6.
>
> | *Dataset*   | *TopER-1*         | *TopER-2*         | *TopER-3*         | *TopER-4*         |
> |---------------|---------------------|---------------------|---------------------|---------------------|
> | *BZR*       | 82.48 ± 1.98        | 84.70 ± 2.84        | 85.66 ± 5.00        | 86.68 ± 3.81        |
> | *COX2*      | 78.81 ± 1.94        | 79.26 ± 4.86        | 79.04 ± 7.49        | 80.30 ± 3.91        |
> | *MUTAG*     | 86.14 ± 6.38        | 88.33 ± 3.88        | 86.75 ± 4.78        | 88.30 ± 4.63        |
> | *PROTEINS*  | 74.03 ± 2.71        | 74.67 ± 2.73        | 75.21 ± 3.39        | 75.65 ± 3.87        |
> | *IMDB-B*    | 73.00 ± 4.40        | 74.20 ± 4.26        | 74.50 ± 3.50        | 74.70 ± 3.95        |
> | *IMDB-M*    | 48.73 ± 4.33        | 49.80 ± 2.94        | 49.73 ± 4.18        | 49.87 ± 4.00        |
> | *REDDIT-B*  | 81.95 ± 2.74        | 90.45 ± 2.55        | 91.05 ± 2.62        | 91.50 ± 2.01        |
> | *REDDIT-5K* | 50.21 ± 1.41        | 54.11 ± 2.43        | 56.19 ± 2.40        | 56.33 ± 2.74        |
>
> **W4. Contribution of Each filtration:**
> >The ablation study does not analyze the effects of each filtration function on individual datasets, missing insights into how specific functions influence results across different types of data.
>
> Thank you for your insightful feedback. We would like to draw your attention to Table 6, which presents the individual performance of each filtration function on each dataset. This table aims to provide a detailed analysis of specific filtration functions' performances across all datasets. For the synergetic contribution of each filtration function, we refer the table in our previous response. If there are specific aspects you believe we have overlooked, we would be happy to provide further clarification or expand on this analysis in the revision.
>
> *Thank you once again for your valuable feedback. We are happy to address any further questions or comments you may have.*

---

> > ### Comment · Reviewer_4YJQ · 2024-11-25
> >
> > Thank you for your in depth response to my comments. My questions have all been adequately answered. Therefore, I  will raise my score.

---

> > > ### Author Response · Authors · 2024-11-25
> > >
> > > Thank you for raising your score. We truly appreciate your constructive and insightful feedback, which has helped us improve our submission.

---

### Meta-Review · Area_Chair_29hL · 2024-12-19

**Metareview:**

This work introduces TopER, an approach to graph embedding that incorporates topological data analysis (TDA) principles to produce low-dimensional. By simplifying Persistent Homology and focusing on the evolution of graph substructures through filtration and linear regression, TopER demonstrates competitive performance in clustering and classification tasks across various datasets. The experimental results support the effectiveness of TopER in generating self-supervised graph representations, highlighting its ability to characterize graph topological structures using only two linear parameters (a and b).

However, I have concerns that this work may not be suited for a machine learning conference like ICLR. The method relies heavily on hand-designed components and very low dimensional representations, which limits its generality and adaptability to new tasks without heavy engineering. While the results look promising, the approach seems to be more of an application of existing TDA techniques to graph representation learning rather than a fundamental contribution to machine learning on graphs. I would like to see a stronger machine learning contribution, with more emphasis on learning-based approaches to the filtration, rather than relying solely on hand-designed features and linear regression.

**Additional Comments On Reviewer Discussion:**

I read the paper and it seems below bar. Reviewers were far too positive, while not providing a compelling reason to accept the paper. Given the borderline status, I recommend rejection.

---

### Decision · Program_Chairs · 2025-01-22

Reject